# DeepISMNet: Three-Dimensional Implicit Structural Modeling with Convolutional Neural Network

Zhengfa Bi[1], Xinming Wu[1], Zhaoliang Li[2], Dekuan Chang[3], and Xueshan Yong[3]

[1]School of Earth and Space Sciences, University of Science and Technology of China, Hefei, Anhui, P.R.China.
[2]China Aero Geophysical Survey and Remote Sensing Center for Natural Resources, Beijing, P.R.China.
[3]Research Institute of Petroleum Exploration & Development-NorthWest(NWGI), PetroChina, Gansu, Lanzhou, P.R.China.

**Correspondence:** Xinming Wu (xinmwu@ustc.edu.cn)

**Abstract.** Implicit structural modeling using sparse and unevenly distributed data is essential for various scientific and societal purposes ranging from natural source exploration to geological hazard forecasts. Most advanced implicit approaches formulate structural modeling as least-squares minimization or spatial interpolation, using various mathematical methods to solve for a scalar field that optimally fits all the inputs under an assumption of smooth regularization. However, these approaches may not reasonably represent complex geometries and relationships of structures and may fail to fit a global structural trend when the known data are too sparse or unevenly distributed. Additionally, solving a large system of mathematical equations with iterative optimization solvers could be computationally expensive in 3-D. To deal with these issues, we propose an efficient deep learning method using a convolution neural network to create a full structural model from the sparse interpretations of stratigraphic interfaces and faults. The network is beneficial for the flexible incorporation of geological empirical knowledge when trained by numerous synthetic models with realistic structures that are automatically generated from a data simulation workflow. It also presents an impressive characteristic of integrating various types of geological constraints by optimally minimizing a hybrid loss function in training, opening new opportunities for further improving the structural modeling performance. Moreover, the deep neural network, after training, is highly efficient to generate structural models in many geological applications. The capacity of our approach for modeling complexly deformed structures is demonstrated by using both synthetic and field datasets, in which the produced models can be geologically reasonable and structurally consistent with the inputs.

## 1 Introduction

A geological model structurally consistent with the subsurface is essential for well understand the subsurface spatial organization and quantitatively simulating geological processes for a wide variety of earth science applications (Li et al., 2016; Wellmann and Caumon, 2018). Structural modeling is aimed to accurately represent the geometry of geological structures with a numerical model by using various mathematical methods. The traditional modeling approach can be described as explicit or surface modeling (Caumon et al., 2009). It reproduces the complex geometries and relationships of structures by digitizing the interpreted surface elements and their arrangements, and the resultant model typically incorporates a series of geological interfaces derived by a triangulation algorithm. In addition to being time-consuming, the modeling process is also related to each individual geologist's interpretations and might not be replicated by others (Caumon et al., 2009; Chaodong et al., 2010).

Recently, more and more implicit structural modeling methods have been proposed for constructing geological models because of their efficient, updatable, and reproducible characteristics (Calcagno et al., 2008; Caumon et al., 2012; Hillier et al., 2014; Laurent et al., 2014; Collon et al., 2015). The implicit method is distinguished from the explicit approach because it consists in interpolating field structural observations into a volumetric scalar function that is defined on the entire region of interest to implicitly represent geological structures. In this function, the geological interfaces are embedded as its iso-

surfaces, while the structural discontinuities are indicated by discontinuous value jumps of the function. Thus, the scalar function is also called the implicit model. The implicit method benefits from incorporating all available geological information into the resultant model by integrating the observed data and the empirical knowledge, providing an effective alternative to reproduce the geometry of the subsurface from a global view (Calcagno et al., 2008; Fossen, 2016). The input structural data of the implicit method typically included various types of modeling objects, such as spatial points, vectors, polylines, and

surfaces, interpreted from field observations. The empirical knowledge can be manually inferred from the structural data by the geologists and geophysicists to define the possible geometrical relationships among the geological interfaces and drive the modeling behaviors of the implicit methods. The output model requires representing geologically reasonable structures while honoring the input structural data. As it is hardly possible to observe the ground truth of subsurface, the geological structures are often heterogeneously sampled in a limited number of highly developed mining and oil fields. This arises the necessity of

adding prior geological rules and assumptions to constrain the modeling process. For example, the existing implicit interpolants typically impose explicit smoothness criteria to simplify local variations for computing a unique model.

    The discrete smooth interpolation (DSI) is one class of implicit methods that compute structural models by discretizing the scalar function on a volumetric mesh (Mallet, 1988, 1992, 1997, 2014; Souche et al., 2014; Renaudeau et al., 2019). In DSI and its variant approaches, structural modeling is performed by solving a least-squares minimization problem with smooth

constraints to compute a scalar field compatible with the inputs. This smooth constraint incorporates empirical geological knowledge into the modeling process with a fundamental assumption that the desired model should be as smooth as possible. However, the mesh elements prohibit from crossing structural discontinuities because the scalar function is always continuous on the mesh elements, and the method cannot correctly estimate the gradients of the scalar function near the faults or unconformities (Shewchuk, 2002). To deal with this problem, we require to compute a constrained unstructured mesh by in-

dependently modeling the discontinuous structures, such that the approaches can work well in these cases. In addition to DSI, the potential field method (PFM) is another class of implicit approaches (Lajaunie et al., 1997; Jessell, 2001; McInerney et al., 2007; Phillips et al., 2007). PFM typically formulates structural modeling as a dual cokriging interpolation (Chiles et al., 2004; Calcagno et al., 2008) or as a radial basis function interpolation (Carr et al., 2001). In comparison to DSI, although the models are evaluated on a volumetric mesh for a visual purpose, PFM does not use any mesh grids when computing the scalar function.

Instead, structure interpolation is fully dependent on the distribution of the observed structural data and the influence range of each data point is determined by the chosen interpolants. However, PFM usually yields a dense system to scale the influence of the interpolants over the entire volume for obtaining a structurally valid solution, which causes the computational cost quickly increase with the input data size and soon become prohibitive.

The existing approaches exhibit many promising characteristics, however, reproducing structures of highly deformed regions remains a challenging task regarding geological consistency because the modeling reliability depends on the availability and quality of the observed data. Structural interpolation fully guided by mathematical equations might not always produce a geologically valid model given sparse or unevenly sampled data (sparse or clustered) in some complex geological circumstances. Corresponding structural models often have erroneous geometries that are incompatible with geological knowledge and spatial relationships with relevant structures. This problem is mainly attributed to the limited constraints that are permitted in structural interpolants, in which all the data and knowledge are mathematically represented as a form of linear constraints to compute a continuous scalar function as smoothly as possible. Although this assumption is helpful to derive a unique model, imposing such a smoothness criterion might compromise the influence of local structural variations and negatively impact the modeling accuracy of highly variant structures (de Kemp et al., 2017; Hillier et al., 2021). Because the modeling flexibility is limited to the models that a specific interpolant can generate, the implicit methods usually suffer from artefacts or geometrical features physically impossible from a geological modeling point of view. Therefore, it is significant to improve implicit modeling by flexibly aggregating all available geological information to ensure that we obtain a structurally reasonable model (Grose et al., 2018, 2021).

In this study, we present a deep learning method using a convolutional neural network (CNN) as an alternative to conventional implicit structural modeling. Deep learning is a type of data-driven and statistical approach that estimate an implicit function that maps inputs to outputs from past experiences or example data by minimizing given quality criteria (Donmez, 2010). In contrast with traditional approaches, deep learning is beneficial for making a prediction without solving a linear system of equations under prior mathematical constraints at cost of expensive computation. Among current learning-based methods, CNN is essential for its remarkable power in analyzing geometrical features and capturing complexly nonlinear mapping relations between inputs and outputs given a sufficiently large training dataset. To find an optimal trade-off between accuracy and efficiency, there exist many convolutional modules available for constructing the CNN architecture, such as depth-wise separable convolution (Chollet, 2017; Howard et al., 2017), attention mechanism (Iandola et al., 2016; Howard et al., 2019), and residual learning structure (Sandler et al., 2018). It is not a surprise that the CNN-related applications in geosciences have been growing rapidly during the past years, including seismic interpretation (Shi et al., 2019; Wu et al., 2019; Geng et al., 2020; Bi et al., 2021), earthquake detection and location (Wu et al., 2018; Perol et al., 2018), remote-sensing image classification (Chen et al., 2016; Maggiori et al., 2016), geochemical map interpolation (Kirkwood et al., 2022), and so on. It is worth noting that a novel learning-based method using Graph Neural Networks (GNN) (Hillier et al., 2021) has been recently developed to integrate structural observations into a graphic mesh encoding all relevant geometrical relations for producing a structural scalar field. This method presents a promising foundation for introducing interpolation constraints that current implicit mathematical methods cannot permit when comparing the prediction and the structural observations, showing an impressive performance to deal with implicit and discrete data. However, the method cannot exactly reproduce the modeling results under the same inputs as the network parameters are initialized randomly in each run of computation. By measuring structural errors only on the scalar field constraints, it may fail to incorporate information associated with structural discontinuities into graphic structures, such as representing the spatial relation of the modeling elements across faults. Another potential limitation results from a

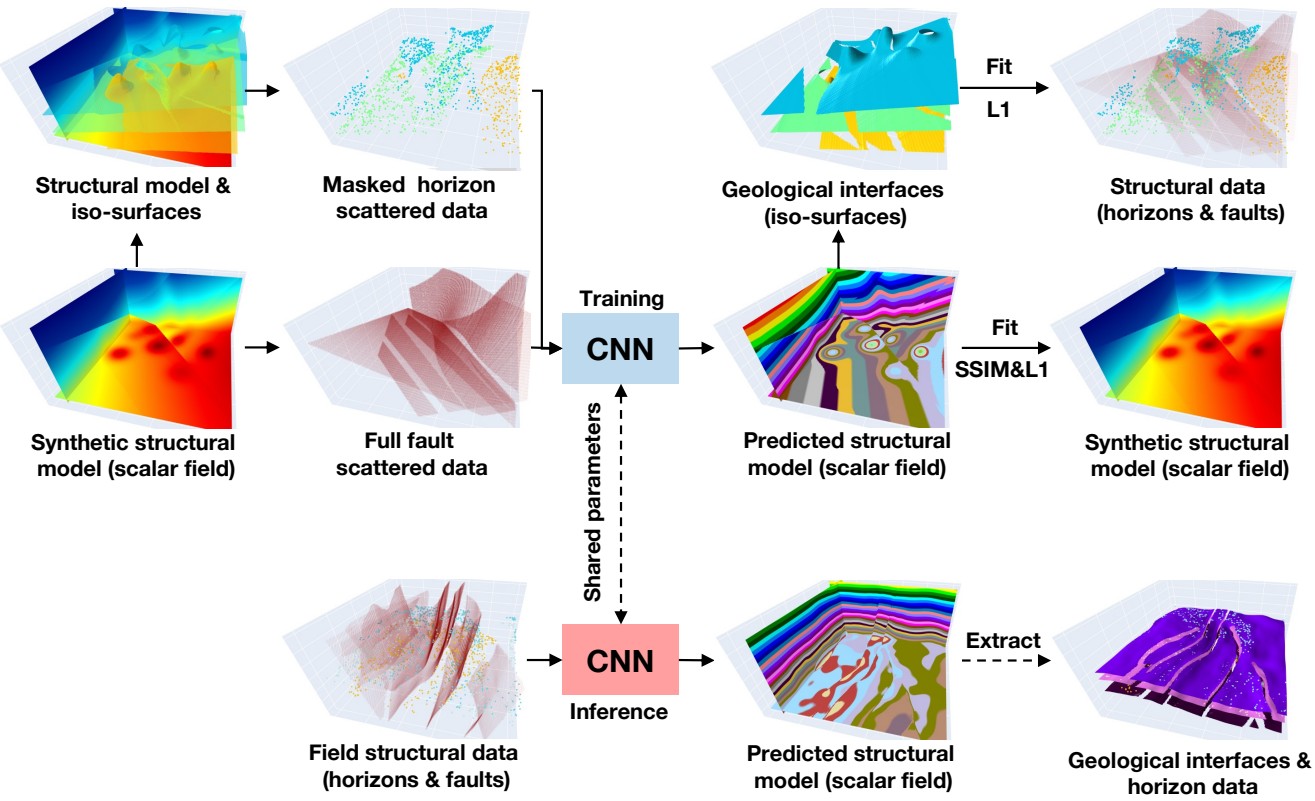

**Figure 1.** Our implicit modeling method produces a volumetric scalar function as an implicit representation of all the geological structures from input structural data by using CNN. Trained with numerous synthetic data, the network can be applied to field structural data to efficiently predict a geologically reasonable model that well matches the inputs.

bottle-necking issue in the current GNN's architecture (Alon and Yahav, 2020), in which further improvement of the modeling
capacity is restrained by network depth with a few layers. A network with a simple structure might not be sufficient to deal with relatively complex geological structures.

As is shown in Figure 1, we formulate implicit modeling as an image inpainting problem with deep learning, in which a full structural model is estimated from the sparse and heterogeneously sampled data based on the past experiences and knowledge learned from training dataset. This characteristic permits a flexible introduction of empirical geometrical relations and
structural interpolation constraints by defining an appropriate loss function to measure the differences between the structural models being compared. Our network, also called DeepISMNet, can produce a scalar field as an implicit representation of all the structures from various types of geological data that includes horizons and faults to encode the stratigraphic sequence and control the geological boundaries, respectively. We parameterize faulting and folding simulations to automatically create numerous structural models with realistic and diverse structures by randomly choosing parameters within reasonable ranges,
which are considered as the example data or labels. In training the network, we randomly extract horizon and fault structures

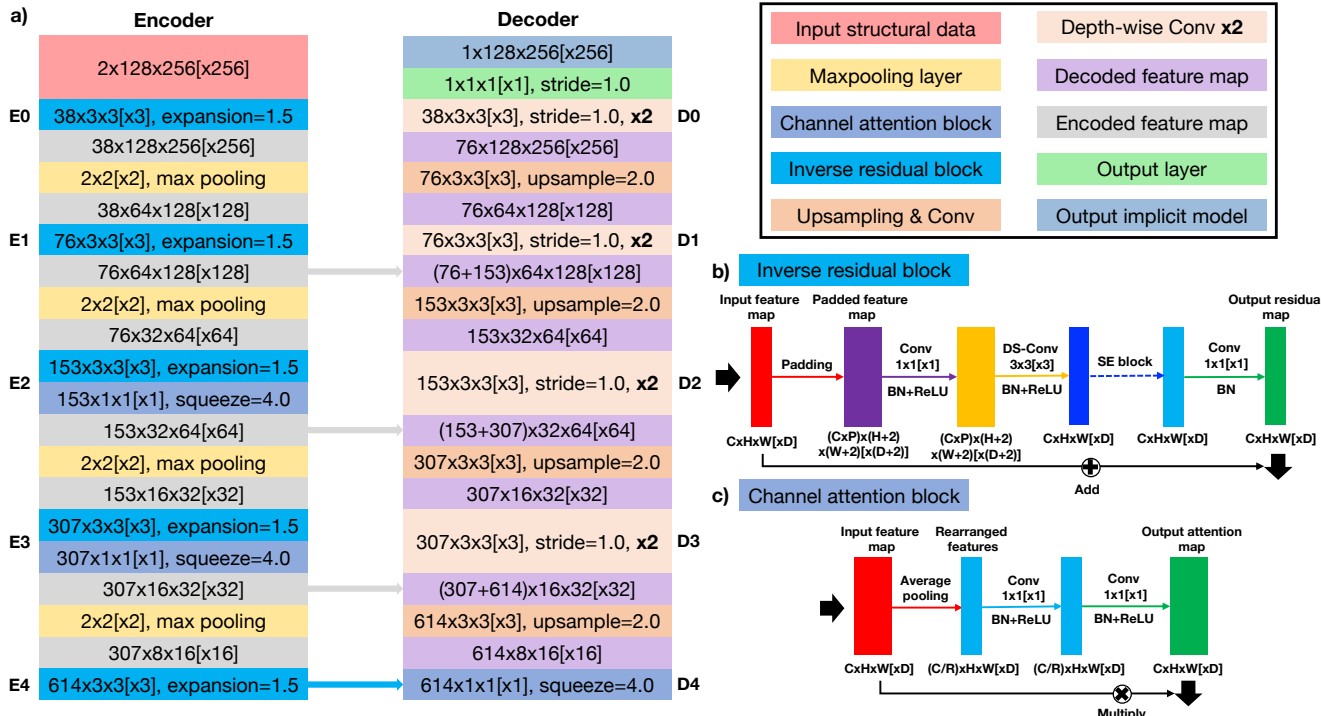

**Figure 2.** Our network has a U-shaped architecture that consists of encoder and decoder branches shown in (a). The encoder uses an inverse residual block (b) supplemented by a lightweight channel-based attention (c) to deal with the input structural data at each of the 5 different spatial scales. The decoder computes the hidden representations at the corresponding 5 resolution scales to form a sufficiently deep CNN. Note that square brackets represent the dimensional expansion of the corresponding 2-D networks to 3-D ones.

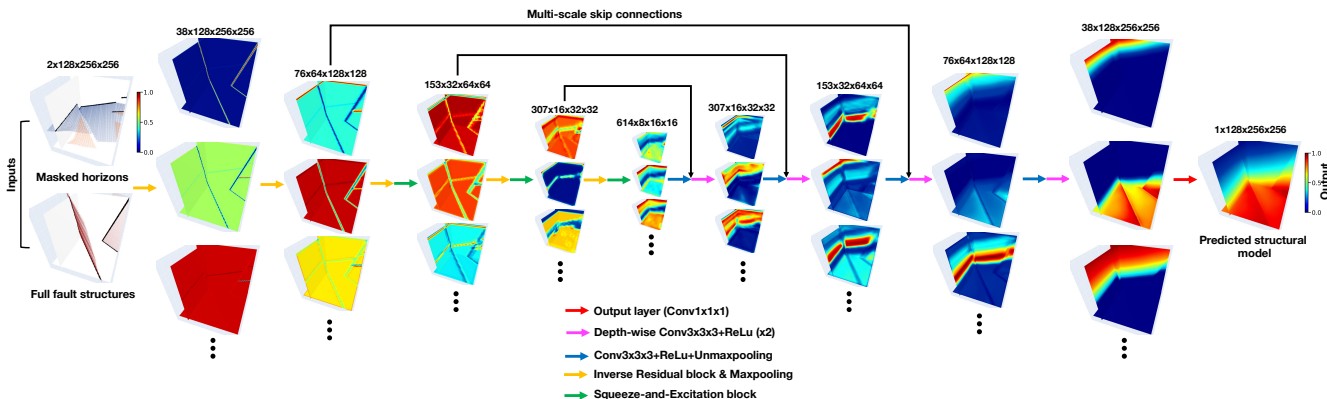

**Figure 3.** The normalized hidden feature representations computed in each spatial scale of the 3-D structural modeling networks.

from these models to further generate unevenly distributed data as inputs that the network takes to predict a full geological model as output. Also, we demonstrate that the normal vectors sampled near geological interfaces can be used to constrain local structural orientations associated with the gradients of the predicted model. In training the CNN, we define a hybrid loss function that combines element-wisely measurement on the input horizon data and multi-scale structural similarity over the local sliding windows to guarantee a geologically compatible prediction. Once obtaining a structural model, we can simply compute the horizon surfaces by using the iso-surface extraction method while detecting the faults near the local value jumps or discontinuities (Figure 1). We find that the trained CNN can efficiently create a geologically reasonable and structurally consistent model in both synthetic and field data applications, showing promising potential for further leveraging deep learning to improve modeling capacity in many geological applications. In addition, the solutions can be reproducible as it is not necessary to randomly initialize the parameters of the trained network at each modeling process.

We organize this paper as follows. In the Methodology section, we describe the CNN architecture designed for implicit modeling and its associated loss function definition. In the Data Preparation section, we introduce the methodology used to automatically generates training samples and simulate the partially missing horizons. The Implementation and Application sections include both synthetic and real-world case studies to verify the performance of our network in representing complex geological structures. The Discussion section presents the promising characteristics of our CNN approach and its current limitations and possible improvements that we will focus on in future research. Finally, we summarize the work in the Conclusions section.

## 2    METHODOLOGY

In this section, we describe the CNN architecture and its associated loss function used in training the network to generate implicit structural models.

### 2.1    Network Architecture

Our developed CNN architecture uses a U-shaped framework modified from UNet and its associated variants (Ronneberger et al., 2015; Zhou et al., 2018), where we include further improvements based on previous works to find an optimal trade-off between accuracy and efficiency in geological modeling. In many image recognition tasks where inputs and outputs share the same spatial resolution, UNet is typically regarded as a standard principle due to its excellent performance (Lin et al., 2017; Yu et al., 2018a). Its great representational power results from a linked encoder-decoder architecture, in which features are first downsampled at multiple spatial resolutions in the encoder and then recombined with their upsampled counterparts through skip connections in the decoder. The localized components of the inputs are typically extracted at an early stage of the CNN, while the relatively high-level and global features are obtained when the receptive fields are increasingly large in deep convolutional layers. Thus, as the hidden representations with different spatial resolutions have much distinctive geometrical information, systematically aggregating the hierarchical multi-scale features with skip connections is attributed to a reliable and stable structural field prediction. Furthermore, the low-level features computed from the shallow layers better follow the

input structures than the deep high-level features because the structural information might be gradually missing in recursive feature compressions at the downsampled spatial resolutions. The use of skip connections also helps to enhance the low-level features and produce a model structurally consistent with the inputs.

We show the proposed 2-D and 3-D CNNs with the same architecture in Figure 2a, in which square brackets represent the dimensional expansion of the corresponding 2-D networks to 3-D ones. The encoder branch in our proposed network consists of 5 successive inverse residual blocks dealing with the input structural data at 5 different spatial scales (from $\mathbf{E}0$ to $\mathbf{E}4$) related to 2, 4, 8, 16, and 32 downsampling rates, respectively. When downsampling the hidden representations by using the max pool layer, the encoder simultaneously expands its channels at each spatial scale. As is shown in Figure 2b, we adopt a linear bottleneck and inverted residual architecture in each block to make an efficient convolutional structure by leveraging the low-rank nature of the inputs. This structure is composed of a $1 \times 1[\times 1]$ expansion convolutional layer, a $3 \times 3[\times 3]$ depth-wise convolutional layer, and another $1 \times 1[\times 1]$ projection convolutional layer, and each convolution is followed by a Batch Normalization (BN) and a Rectified Linear Unit (ReLu). The two $1 \times 1[\times 1]$ convolutional layers at the ends of the depth-wise convolutional layer are designed to expand the inputs to higher-dimensional feature space (one and a half times of the channels) and project them back to the output channels, such that the inverse residual block forms a compact feature embedding to improve the expressiveness of the nonlinear transformation at each channel. We did not try a larger expansion factor because of the GPU memory limitation, but we would suggest choosing a larger size if the GPU memory is allowed. With a residual connection over the expansion and projection convolutional layers, the block is formulated as a residual learning function to speed up the backpropagation of gradient responses. Although the encoder layers aggregate abundant information through recursive channel expansions, not all the features are useful for modeling structures. There exist many structurally irrelevant features with mostly zeros across channels because of the sparse and heterogeneous characteristics of the inputs. Treating all channel-wise features equally would waste unnecessary computations to focus on the informative features and thus negatively influences the representational power of the network. To enhance the CNN's discriminative learning ability, we insert a lightweight channel-based attention module into the bottleneck structure of the inverse residual block in the last three spatial scales of the encoder. The attention block (Figure 2c) consists of squeeze and citation modules, in which the input features are first compressed into lower-dimensional feature space in the squeeze module and then transformed to the channel-wise attention weights with the same channels as the inputs in the citation module. This module (Hu et al., 2018) encourages the network to adaptively learn the relations across hundreds of high-level features with relatively global structural information and rescale their importance to stabilize the modeling by emphasizing the informative features and suppressing the irrelevant ones.

The decoder branch includes the 5 spatial scales (from $\mathbf{D}4$ to $\mathbf{D}0$) consistent with the encoder to form a sufficiently deep network and large receptive field for structural interpolating. It is responsible for integrating the hidden representations from the previous unmaxpooling layers and the encoder skip connections while compensating for the spatial resolution mismatch between the concatenated features. In each spatial scale, the upsampled decoded features are concatenated with their downsampled counterparts from the encoder branch and further sequentially refined in the two convolutional layers. We use depth-wise separable convolutions (Howard et al., 2017) as an efficient replacement for the traditional convolutions. The depth-wise sep-

arable convolutional layer factorizes the convolutional operation as two separate layers including a lightweight $3 \times 3[\times 3]$ convolutional layer for filtering features within each channel and a relatively heavy $1 \times 1[\times 1]$ convolutional layer for combining features across channels. By splitting the standard convolution as the two-step process, we can dramatically reduce the computation complexity and the GPU memory to construct a lightweight decoder network. In the output layer, we adopt a simple linear transform that is implemented by a $1 \times 1[\times 1]$ convolutional layer to cross all the decoded features for finally producing a structural field.

Figure 3 visualizes the normalized hidden representations at each resolution scale of our 3-D structural modeling network that the inputs are passed through. As the amplitude ranges of the hidden representations are much varying from each other, we rescale them to obtain the normalized features with values restrained from zero to one for a visual purpose. Our CNN is designed to progressively complete structural features layer by layer through sequential non-linear convolutional units that are conditioned on the previous convolutions. As is displayed in Figure 3, the valid convolutional responses only exist near the input structures in the starting layer of the network. To spread geological structures elsewhere, every convolutional layer collects information from the previous layer outputs within an increasingly expanding receptive region by staking convolutions and recursively downsampling the input hidden features. Therefore, at the bottom layers of the network, the structural information in the inputs can be used to constrain the modeling process over the entire model from a global receptive region of view. This characteristic allows the network to correctly understand the relations of the spatially distant but contextually close features. Although weighting on spatial proximity is typically used in many traditional structural interpolation methods, the nearby features are not necessarily more significant than distinct ones for making a geological-related prediction. For example, when the stratigraphic layers are located the opposite of a large shear zone or other discontinuous structures, the correlations of distinct data points computed from a global review are essential to capture a more accurate structural pattern.

### 2.1.1 Loss Function

The network provides an attractive characteristic to integrate various structural constraints by minimizing the corresponding errors between the predicted and reference models. For making geologically valid predictions, we combine element-wise accuracy with multi-scale structural similarity to define a hybrid loss function. We introduce notations and formal definitions used in this loss function. Let $\mathbf{x}$ be reference structural model, and $\mathbf{m}$ be its binary mask where the pixels or voxels on the input horizon data are set to $1$ and the rests are set to $0$. The dimensional sizes of $\mathbf{x}$ and $\mathbf{m}$ are consistent with the samples in our training dataset. For each reference model $\mathbf{x}$, the CNN $f_\theta$ with trainable parameters $\theta$ takes horizon data $\mathbf{h} = \mathbf{x} \odot \mathbf{m}$ and fault data $\mathbf{f}$ to create a structural field $\hat{\mathbf{y}} = f_\theta(\mathbf{h}, \mathbf{f})$ as outputs. We denote the predicted model that is replaced with the inputs on the horizon data as $\mathbf{y} = \hat{\mathbf{y}} \odot (1 - \mathbf{m}) + \mathbf{x} \odot \mathbf{m}$.

In many geologically related regression problems, Mean Square Error (MSE) and Mean Absolute Error (MAE) are commonly used to element-wisely measure the accuracy of the solutions (Geng et al., 2020; Hillier et al., 2021). However, MSE typically emphasizes the elements with larger errors but is more tolerant of smaller ones, regardless of the underlying spatial pattern of the data. In comparison with MSE, MAE is more sensitive to the local structural variations, reducing the artefacts

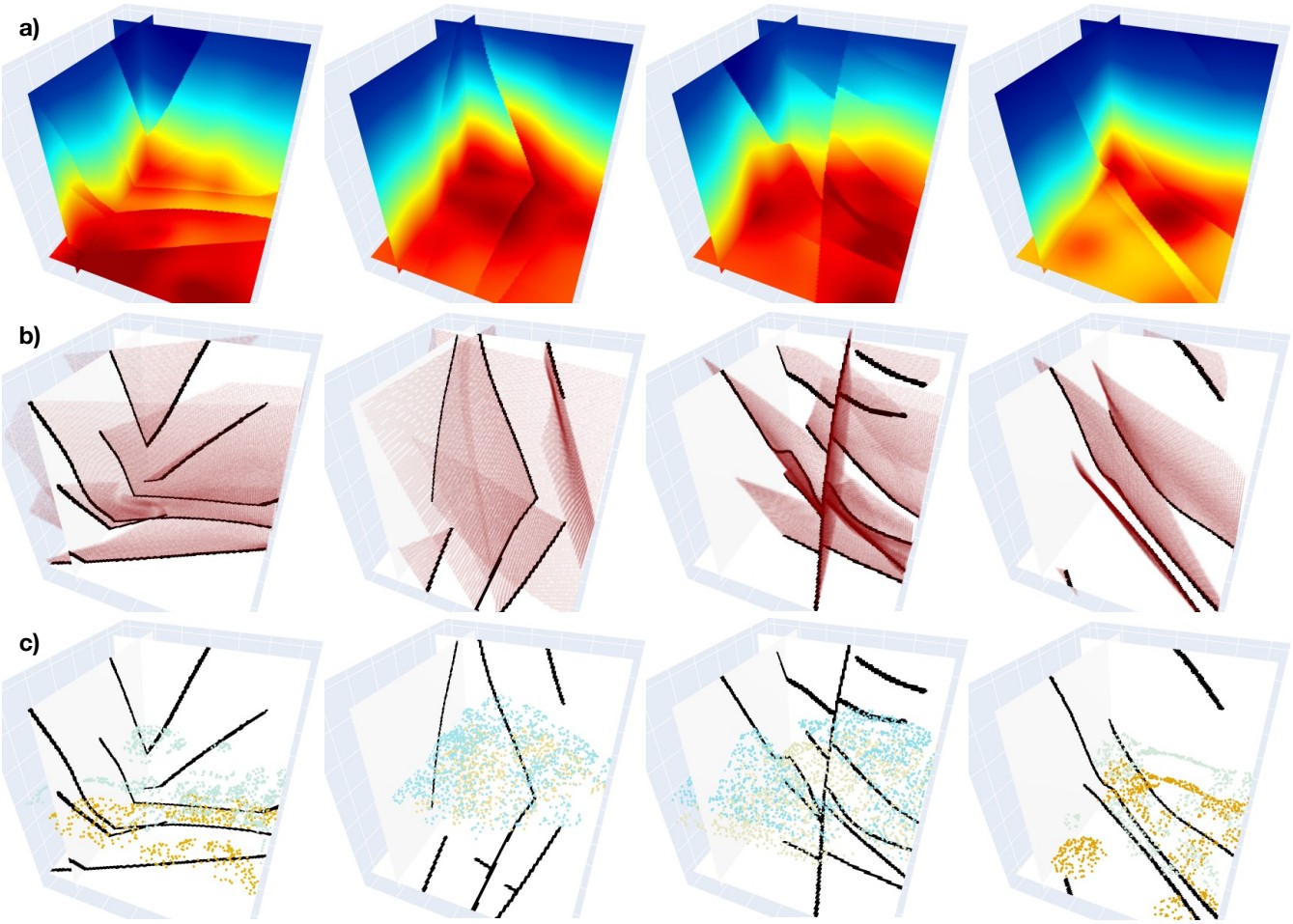

**Figure 4.** Four pairs of 3-D training data samples. The first row shows 3-D synthetic implicit structural models used as labels in training our 3-D network. The second and third rows, respectively, display the fault and sparse horizon points extracted from the models (first row), which are together used as inputs of the CNN.

caused by excessively penalizing large errors. We adopt masked MAE as a point-wise measurement in the hybrid loss function, which is formulated as follows:

$$\mathcal{L}_{\text{mae}}(\mathbf{p}) = \frac{1}{N} \sum_{p \in \mathbf{p}} |\mathbf{x}(p) - \hat{\mathbf{y}}(p)|, \tag{1}$$

where $N$ represents the total number of the points within a square patch $\mathbf{p}$. We crop the patches from the same spatial location in the two structural models being compared.

Although MAE outperforms MSE in geological modeling scenarios, the results are still not optimal. CNN trained by using MAE alone might not correctly capture geometrical features that are represented by the distribution of the neighboring points,

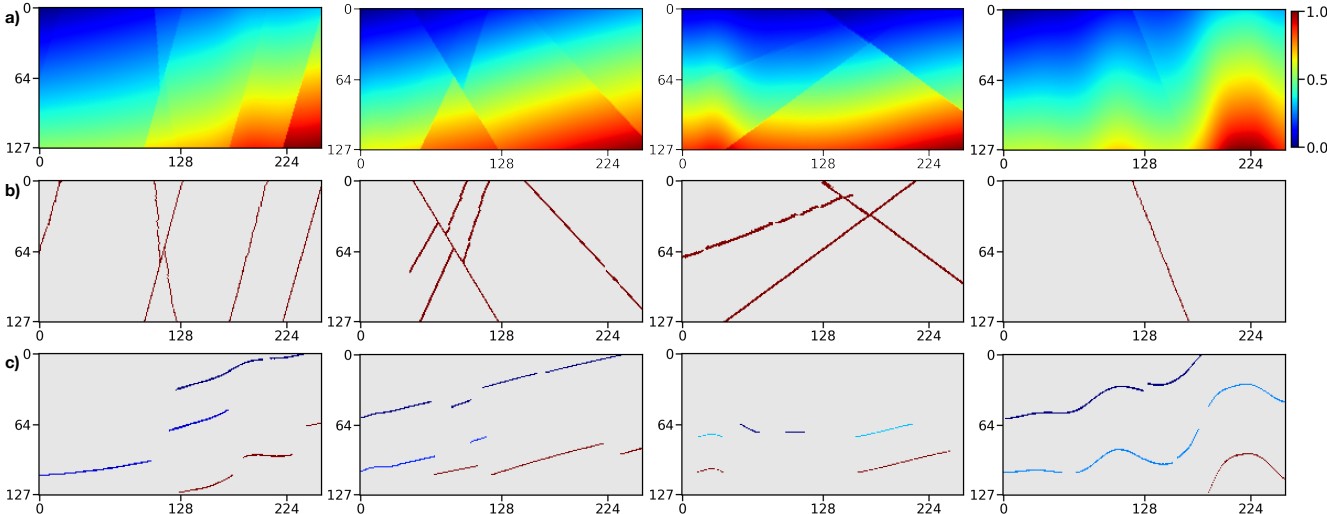

**Figure 5.** Four pairs of 2-D training data samples. The first row displays 2-D synthetic implicit structural models used as labels in training our 2-D CNN. The second and third rows, respectively, show the fault images and sparse horizon points extracted from the label models (first row), which are together used as inputs if the CNN. It is worth noting that the points denoted by the same color in each image of the third row correspond to the same horizon.

whereas blurring high-frequency and sharp structural discontinuities. Thus, the two models with similar MAE might appear significantly distinct structures, which negatively impacts the optimization of the CNN's parameters. To alleviate such smooth

effectiveness, we use a hybrid loss function by combining MAE with Structural Similarity (SSIM) (Wang et al., 2003, 2004; Zhao et al., 2016). By adaptively assigning higher weights to the structural boundaries in which the structures present significant contrasts, SSIM can better preserve the high-frequency geometrical features. SSIM loss measuring the CNN prediction and the reference model within the patch $\mathbf{p}$ can be represented as follows,

$$
\begin{aligned}
\mathcal{L}_{\text{ssim}}(\mathbf{p}) &= 1 - (\frac{2\mu_{\mathbf{x}}\mu_{\mathbf{y}} + C_1}{\mu_{\mathbf{x}}^2 + \mu_{\mathbf{y}}^2 + C_1})^{\beta}(\frac{2\sigma_{\mathbf{xy}} + C_2}{\sigma_{\mathbf{x}}^2 + \sigma_{\mathbf{y}}^2 + C_2})^{\gamma} \\
&= 1 - l(\mathbf{p})^{\beta} \cdot cs(\mathbf{p})^{\gamma},
\end{aligned}
\tag{2}
$$

where $\mu_{\mathbf{x}}$ and $\mu_{\mathbf{y}}$ represent the means of model $\mathbf{x}$ and $\mathbf{y}$ within $\mathbf{p}$, respectively. $\sigma_{\mathbf{x}}$ and $\sigma_{\mathbf{y}}$ are the variances, while $\sigma_{\mathbf{xy}}$ is the covariance of the two patches being measured. The means, variances, and covariance are computed by using an isotropic Gaussian filter $G_{\sigma_g}$ with standard deviation $\sigma_g$ and zero mean. Approximately, $\mu_{\mathbf{x}}$ and $\sigma_{\mathbf{x}}$ can be viewed as estimates of the stratigraphic sequences and structural variations in a local patch of model $\mathbf{x}$, and $\sigma_{\mathbf{xy}}$ measures the tendency of the patches in model $\mathbf{x}$ and $\mathbf{y}$ to vary together, thus an indication of structural similarity. $\beta$ and $\gamma$ define the relative significance of the two

terms $l$ and $cs$, which are both set to 1 based on Wang et al. (2003). In addition, we use two small constant factors $C_1$ and $C_2$ to avoid the numerically unstable circumstance of dividing by zero.

The standard deviation $\sigma_g$ of the Gaussian filter $G_{\sigma_g}$ is a hyper-parameter that requires to be defined before training. However, the choice of $\sigma_g$ can impact the prediction accuracy of the network trained by using SSIM. The network trained by SSIM with a large standard deviation $G_{\sigma_g}$ might overly emphasize the local variations and generate spurious features in the prox-
imity of edges while blurring sharp structural boundaries for a small standard deviation $G_{\sigma_g}$ (Zhao et al., 2016). Instead of fine-tuning the parameter $G_{\sigma_g}$, we use Multi-scale Structural Similarity (MS-SSIM) (Wang et al., 2003, 2004) with a dyadic pyramid of $S$ scale levels and formulate it as follows,

$$\mathcal{L}_{\text{ms-ssim}}(\mathbf{p}) = 1 - l_S(\mathbf{p})^\beta \cdot \prod_{j=1}^{S} cs_j(\mathbf{p})^{\gamma_j}, \tag{3}$$

in which $\gamma_j$ are parameters to define the relative importance of each scale level in the variance-related scheme $cs_j$. MS-
SSIM computes a pyramid of patches $\mathbf{p}$ with $S$ spatial scales defined by various $\sigma_g$ of the used Gaussian filter $G_{\sigma_g}$. We define 5 different scales of $\sigma_g = \{0.5, 1, 2, 4, 8\}$, and set each to half of the previous one by recursively downsampling the full-resolution patch using $2 \times 2$ average pool layer ($S = 5$). We adopt $\gamma_j = \{0.05, 0.29, 0.3, 0.24, 0.12\}$ to rescale the losses estimated from the 5 scale levels, and make sure the sum of them is equal to 1 for computing the MS-SSIM loss.

MS-SSIM highlights structural variations focusing on a neighborhood of point $p$ as large as the given Gaussian filter $G_{\sigma_g}$, but
might produce artefacts in the predictions because its derivatives cannot be correctly estimated near the boundary regions of the patch in the optimization. This can be alleviated by supplying an element-wise measurement that is computed on a single point of the patches being compared in the loss function. Additionally, MS-SSIM is not sensitive to uniform biases, which might cause unexpected changes in stratigraphic sequences or shifts of geological interfaces in modeling results. In comparison, although MAE better preserves stratigraphic sequences by minimizing error at each point equally within the patch, it might not
produce quite the same high-frequency contrast as MS-SSIM regardless of local structures. To capture the best characteristics of both loss functions, we thus propose to combine them as,

$$\mathcal{L}_{\text{sum}} = \frac{1}{K} \sum_{i=1}^{K} (\lambda \mathcal{L}_{\text{mae}}(\mathbf{p}_i) + \mathcal{L}_{\text{ms-ssim}}(\mathbf{p}_i)), \tag{4}$$

where $\lambda$ is a weighting factor used to balance the relative importance of different losses, and $K$ represents the number of cropped patches. In training the CNN, we point-wisely crop square patches from the structural models being measured and
compute the loss within each patch based on Equation 4, in which we empirically set the dimensional size of each patch to 7 and the $\lambda$ to 1.25. The total loss $\mathcal{L}_{sum}$ is estimated by averaging the losses computed for the $K$ patches. All the parameters in the loss function are selected based on many numerical experiments and kept fixed throughout the study to avoid the need for tuning. Although we cannot ensure the used parameter combination is the best one, further parameter tuning is much more time-consuming for training a deep CNN but hardly obtains further improvements.

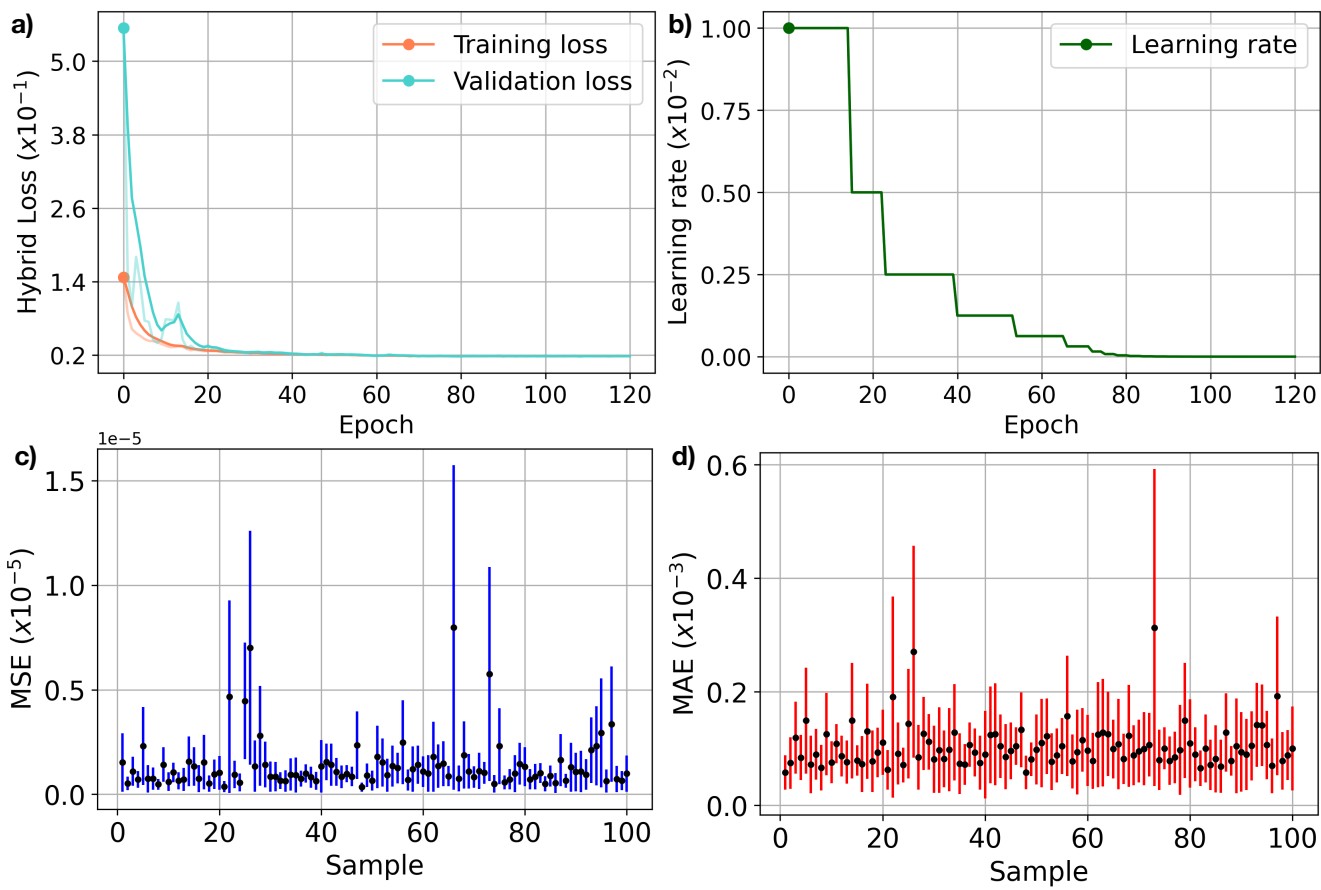

**Figure 6.** Training (cyan) and validation (orange) curves of using our developed hybrid loss (a), and the adaptive adjustment of the learning rate during the training (b). We run 20 times of the trained CNN to compute the MSE (c) and MAE (d) for the 100 models randomly chosen from the validation dataset, in which the input structural data are regenerated in each computation. The black dot represents the average error while the blue and red lines indicate the error ranges of the MSE and MAE, respectively.

## 3  DATA PREPARTION

Our CNN architecture is beneficial for the flexible incorporation of empirical geological knowledge in a supervised learning framework with numerous structural models that are all automatically generated from an automatic data simulation workflow. We randomly delete some segments from the models to obtain the partially missing horizons similar to the modeling objects collected from field observations. In training our network, the incomplete horizons, together with the faults, are used as inputs to predict a structural field under the supervision of the full model.

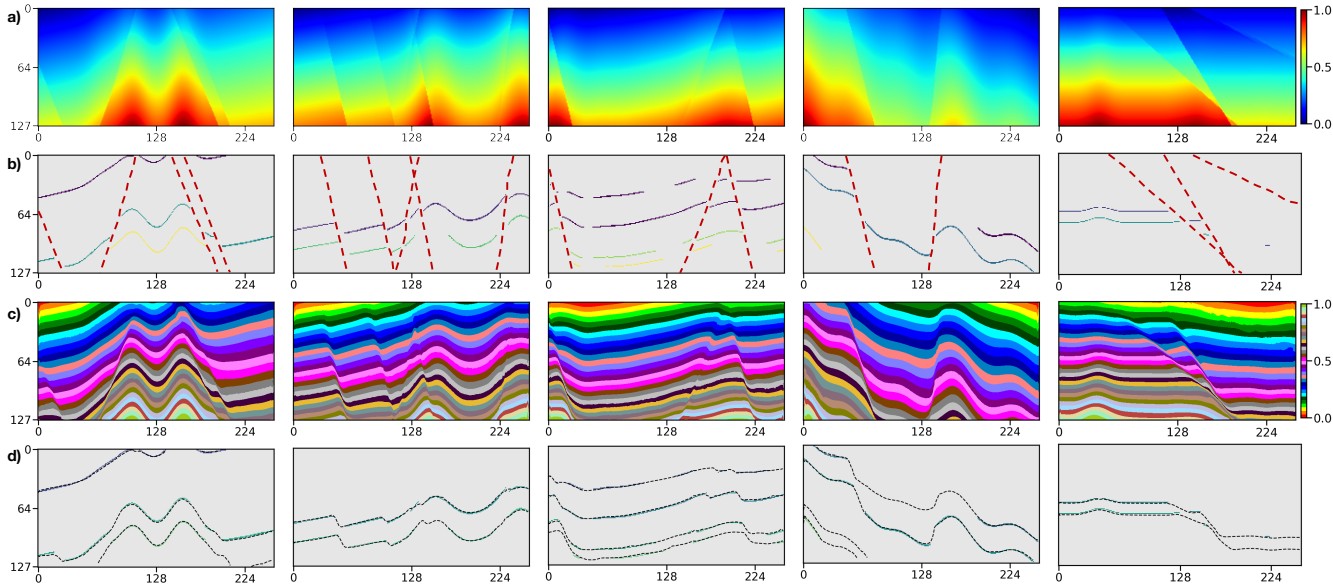

**Figure 7.** We apply the trained CNN to the 5 geological models (a) not included in the training dataset. We randomly generate horizon and fault structural data (b) from the models as the inputs of our network. By visual comparison, the modeling results (c) are nearly identical to the ground truth of the models (a), which can be supported by the great consistency between the scalar field iso-lines (dashed curves) and the input horizons (solid curves) in (d).

## 3.1 Automatic Data Generator

A challenge of applying the supervised learning method is the preparation of many example data and especially the corresponding geological labels for training the network. In the structural modeling, the training dataset should incorporate structurally varying geological models as much as possible to enable the CNN to learn representative knowledge for achieving its reliable generalization in real-world applications. However, it is hardly possible to manually label all geological structures in a field survey because the ground truth of the subsurface is inaccessible. To solve this problem, we use an automatic workflow to simulate geological structural models with some typical folding and faulting features that are controlled by a set of random parameters (Wu et al., 2020). In this workflow, we first create a flat layered model with horizontally constant and vertically monotonically increasing values as an initial model, and then sequentially add folding, dipping, and faulting structures to complicate the features of this model. We simulate the folding and dipping structures by vertically shearing the initial model through a combination of linear and Gaussian-like shift fields, while creating faulting structures by using volumetric vector fields defined around the fault surfaces. By randomly choosing the parameters within the reasonable ranges, we can simulate numerous geological models with diverse and realistic features not limited to a specific pattern. Based on the generated models with known structures, we simply obtain the corresponding ground truth of the fault volume with ones on faults and zeros elsewhere. These geological models can be viewed as volumetric scalar functions because their iso-surfaces represent the cor-

responding stratigraphic interfaces while the local value jumps indicate the structural discontinuities. This is important for our next step of constructing a training dataset to optimize our CNN for geological structural modeling. By using this workflow, we totally obtain 600 3-D structural models shown in Figure 4a, and each model contains $256 \times 256 \times 128$ grid points. These models are vertically flipped up-down and horizontally left-right to build an augmented dataset consisting of a total of $2,400$ pairs of models. At the same time as vertically flipping models, we reverse the sign of their values to ensure that they vertically increase except across a reverse fault. As is shown in Figure 5a, we extract $4$ evenly spaced slices along crossline and inline from the 3-D structural models, respectively, to further build a training dataset for the 2-D network. We use $90\%$ of the datasets for training and the rests for validating the trained network.

## 3.2 Masked Structural Data

The input structural data of our network consists of scattered points that are gridded into a volumetric mesh with valid annotations on structures and zeros elsewhere. As is displayed in Figure 4b and Figure 5b, we label the points near the faults within one pixel to ones and zeros elsewhere to obtain the input fault data. To obtain the input horizon data, we set the points near the horizon surfaces within one pixel to the corresponding iso-values of the structural model, which ensures that the points on the same horizon have consistent annotations. Furthermore, we randomly remove some points from the horizon data in each run of the data generation to simulate the unevenly sampled horizon interpretations.

As the geological interfaces are implicitly embedded in the scalar field with the iso-values and can be obtained by iso-surface extraction methods, we adopt Jittered sampling (Cook, 1986; Hennenfent and Herrmann, 2008) to randomly choose iso-values and obtain the corresponding modeled horizons. This sampling method benefits from remedying the deficiency of the regular sampling method that introduces a specific pattern in the inputs while preserving the beneficial properties of randomness. Specifically, we first sort all the iso-values into a uniformly spaced grid in descending order and then randomly extract one within each grid unit to compute the corresponding horizon. Thus, the horizons extracted from the structural model can be spatially varying and not spaced closely. To remove the points from the horizon data, the simplest method is to randomly generate many square patches and mask the scattered points within the patches. Although this method is commonly used for many image inpainting tasks (Yeh et al., 2017; Yu et al., 2018b), it might negatively impact the CNN to be well generalized in real-world applications for which the inaccessible regions are unlikely in the shape of squares. To solve this issue, we randomize this process by randomly removing points from an individual horizon to prevent the network from learning a specific pattern that all the horizon data are partially missing in the same square regions. Similar to the iso-value selection, we first sort the points on this horizon into a uniformly spaced grid in descending order based on their vertical coordinates, and then randomly mask out the points from one or more grid units. As is displayed in Figure 4c and Figure 5c, the generated partially missing data are similar to the horizons manually interpreted by geologists and geophysicists, which enables the CNN to learn more representative features. Thus, we use this masking method for all the 2-D and 3-D synthetic data experiments in this study.

| 3-D Network architecture | | Computational cost | | Modeling quality metric | | | | | | | |
|---|---|---|---|---|---|---|---|---|---|---|---|
| Name | Backbone | GFLOPs | #Params[MB] | SSIM | EVS | MAE | MSE×10$^{-1}$ | MSLE×10$^{-1}$ | R2S | MDAE | HFA |
| UNet | - | 32.715 | 34.526 | 0.989 | 0.990 | 0.019 | 0.009 | 0.005 | 0.972 | 0.017 | 1.078 |
| AttUNet | - | 33.265 | 34.878 | 0.981 | 0.978 | 0.027 | 0.035 | 0.018 | 0.901 | 0.025 | 1.029 |
| NestUNet | - | 76.406 | 39.091 | 0.839 | 0.773 | 0.129 | 0.250 | 0.126 | 0.288 | 0.115 | 3.025 |
| DeepLabv3$^{+}$ | Xception | 10.328 | 54.510 | 0.988 | 0.990 | 0.194 | 0.008 | 0.004 | 0.977 | 0.017 | 1.634 |
| DeepLabv3$^{+}$ | DRNet54 | 23.293 | 40.672 | 0.989 | 0.991 | 0.019 | 0.008 | 0.004 | 0.978 | 0.018 | 1.228 |
| DeepLabv3$^{+}$ | ResNet101 | 11.042 | 59.226 | 0.986 | 0.985 | 0.024 | 0.014 | 0.007 | 0.956 | 0.022 | 1.623 |
| DeepLabv3$^{+}$ | MobileNetv2 | 4.364 | 7.555 | 0.985 | 0.982 | 0.027 | 0.017 | 0.008 | 0.949 | 0.025 | 1.843 |
| RefineNet | MobileNetv2 | 1.015 | 3.250 | 0.973 | 0.963 | 0.031 | 0.035 | 0.019 | 0.887 | 0.028 | 1.223 |
| RefineNet | MobileNetv3 | **0.937** | **2.600** | 0.977 | 0.981 | 0.030 | 0.022 | 0.011 | 0.937 | 0.028 | 1.735 |
| DeepISMNet* | - | 4.711 | 4.300 | **0.993** | **0.996** | **0.016** | **0.004** | **0.002** | **0.988** | **0.015** | **0.331** |

**Table 1.** A quantitative comparison between our network and the widely used powerful networks using various quality metrics. For each of the quality metrics, the best performance is highlighted in **bold**. The proposed network (DeepISMNet) is marked with an asterisk to distinguish it from the others.

| Loss function | Modeling quality metrics | | | | | | |
|---|---|---|---|---|---|---|---|
| | MAE | MSE×10$^{-1}$ | EVS | R2S | MDAE | SSIM | HFA |
| L1 | 0.017 | 0.005 | 0.994 | 0.986 | 0.016 | 0.991 | 0.527 |
| L2 | 0.017 | 0.005 | 0.995 | 0.987 | 0.016 | 0.989 | 1.321 |
| SmoothL1 | 0.017 | 0.005 | 0.995 | **0.988** | 0.017 | 0.990 | 0.511 |
| SSIM | 0.017 | 0.006 | 0.992 | 0.978 | 0.018 | 0.989 | 0.643 |
| MS-SSIM | 0.017 | 0.005 | 0.994 | 0.986 | 0.016 | 0.991 | 0.630 |
| MS-SSIM&L1 | **0.016** | **0.004** | **0.996** | **0.988** | **0.015** | **0.993** | **0.331** |
| MS-SSIM&L2 | **0.016** | **0.004** | 0.995 | 0.987 | **0.015** | 0.990 | 1.040 |

**Table 2.** A quantitative analysis of our network trained with the distinct loss functions using multiple modeling quality metrics. For each of the quality metrics, the best modeling result is highlighted in **bold**.

## 4  IMPLEMENTATION

In this section, we present the geological structural models derived from our CNN for both synthetic and field data applications to demonstrate its modeling performance.

### 4.1  Training and Validation

Considering the coordinate ranges of the field geological datasets can be much different from each other, we rescale every structural model to obtain the normalized one that ranges from zero to one. This normalization is implemented by first sub-

tracting the minimum and then dividing its maximum and thus would not change its geological structures. When normalizing the structural data, we assign the scattered points on the same geological interface to the corresponding iso-values of the normalized model. In training the network, we formulate these normalized training samples in batches and set the batch size to $4$ based on our computational resources. Within each epoch, the training data are all passed throughout the network to compute the hybrid loss function. We utilize Adam optimization (Kingma and Ba, 2014) with an adaptive learning step length to speed up the network optimization. The initial learning rate is set to $0.01$, which reduces gradually when the criterion performance has stopped further improving. We fold the learning rate by a factor of $0.5$ once the loss stagnates within $2$ iterations. As is shown in Figure 6a, the training and validation loss curves gradually converge to low levels (less than $0.1$) when the optimization stops after $120$ epochs, which demonstrates that the CNN has learned representative geometries and relationships of geological structures from the training dataset. The learning rate is adaptively adjusted as is displayed in Figure 6b during the training.

Furthermore, we evaluate the modeling stability of our network in terms of the perturbations of the input structures created from the same geological model. In this experiment, we randomly choose $100$ synthetic models from the validation dataset and run $20$ times of the trained network to calculate the MSE and MAE for each model. During each modeling process, we randomly generate the horizon scattered points to ensure that the input data are different from each other even for the same structural model. We show the variations of MSE and MAE for each model in Figure 6c and 6d, respectively, in which the MSE and MAE are represented by black dots while the error ranges are denoted by the blue and red lines. We find that most MSE and MAE are less than $0.5 \times 10^{-5}$ and $0.2 \times 10^{-3}$, which are considered to be not very significant in geological modeling scenarios. This demonstrates the proposed CNN architecture is beneficial for implicit structural modeling.

## 4.2  Synthetic Data Examples

When the CNN is well trained, the modeling experiences and knowledge learned from the synthetic dataset are implicitly embedded in the network parameters. To verify its modeling performance, we apply the trained CNN to the $5$ synthetic structural models not included in the training dataset. As is shown in Figure 7a, the models are of complex faulted layered volumes, in which the folded interfaces are reformed by multiple high-angle normal faults. From the original structural models, we generate the incomplete horizon and the fault data (Figure 7b) used as inputs of our network. By visual comparison in Figure 7c, the modeling results with similar geometrical features to the inputs maintain the localized variations of the folded interfaces despite no global structural information used to constrain the model. We further overlap scattered horizon points on the iso-lines of the field predictions in Figure 7d, in which the great consistency between the given structures and the interpolated features again supports our observation in Figure 7c.

As is tabulated in Table 1, the CNN's modeling ability is quantitatively measured by using various quality metrics including SSIM, MSE, MAE, Explained Variance Score (EVS), Mean Squared Log Error (MSLE), Median Absolute Error (MDAE), and R Square Score (R2S) on the entire validation dataset. In addition, we also measure the accuracy of every modeled interface related to the input horizon data by computing Horizon Fitting Error (HFA). This metric measures an average distance between the horizon scattered points and the corresponding iso-surfaces of the predicted model along the vertical axis. Table 1 also shows a quantitative comparison of the proposed method (DeepISMNet) and the other powerful networks commonly used in

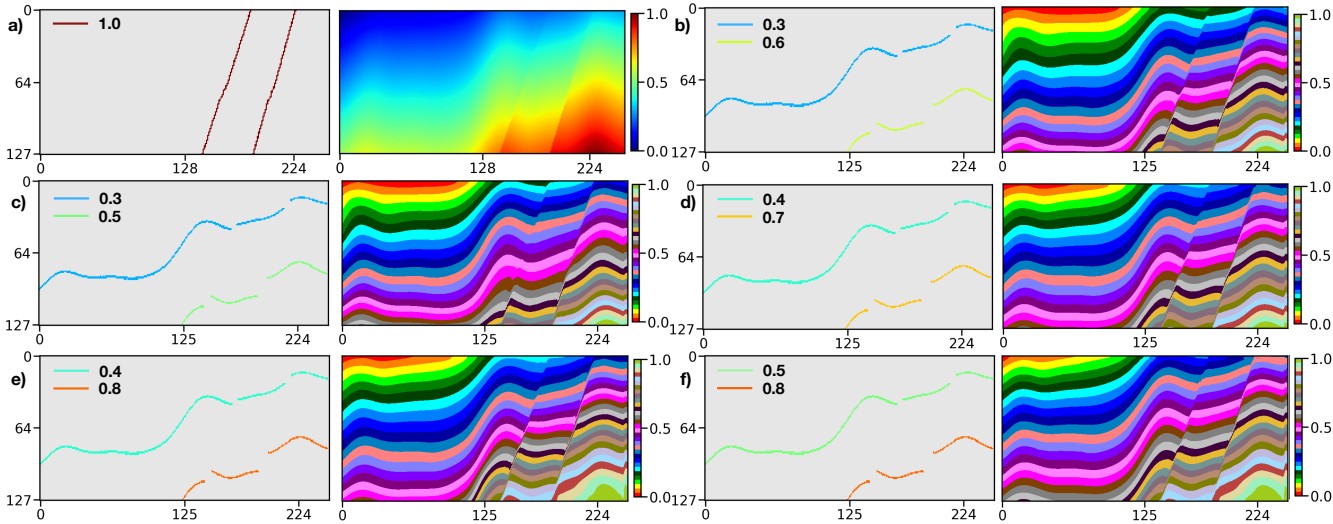

**Figure 8.** We use a synthetic model (a) to study how the varying horizon values impact the CNN's modeling results. As are shown from (b) to (f), the experiments are used to verify that our network can produce almost identical structural models from the same faults and the horizons with different annotations. These indicate that our approach is not sensitive to the changes of the labeled value on each horizon in the inputs, which facilitates its applications in field surveys.

similar scenarios. Our method not only shows better performance on all the metrics but also has a more lightweight architecture with fewer trainable parameters and GFLOPs in comparison to the others. Simplification of the network architecture is mainly associated with the use of inverse residual modules followed by depth-wise separable convolutions in each spatial scale level throughout the network, which enables our CNN to be applied to a large 3-D field modeling task. To guarantee the representational ability of the simplified CNN, the channel-wise dependencies have been explicitly learned by using an attention module that can adaptively highlight more informative features while suppressing irrelevant ones. Therefore, although the number of hidden representations is less than the conventional CNN architectures such as UNet, our network still achieves stable structural interpolation and reliable generalization performance.

## 5 APPLICATION

It might not be surprising that the CNN trained with a synthetic dataset works well to produce a geologically valid and consistent model by using the structural data generated from the same workflow for creating the training dataset. In this section, we further present modeling results of our trained network applied on real-world data that are acquired at different geological surveys to demonstrate proof of concept. The modeling objects collected from field observations or seismic data are required to convert into the uniformly sampling grids to obtain the input structural data of our CNN.

## 5.1   Structural Data Preprocessing

In most cases, the structural data collected from field surveys are discrete and not necessarily located on the sampling grid of the model, such that we need a preprocessing step that scatters the structural data into a volumetric mesh with annotations. To scatter the structural data, we simply shift the horizon and fault interpretations to their nearest sampling grids of the model and obtain the associated scattered points. In both the synthetic and the field data applications, the annotation of fault scattered points is straightforward by simply assigning ones near the faults and zeros elsewhere. However, although the points on horizons can be assigned to the corresponding iso-values of the model in the synthetic data experiments, this might not be feasible when modeling real-world geology from structural interpretations. As the ground truth of geological structures is typically inaccessible before modeling, how to properly annotate the horizon data remains a problem.

We implement a numerical experiment using the horizon data labelled with different iso-values in a synthetic structural model (Figure 8a) to study how they impact the results. In this experiment, the scattered points on two horizons are assigned by the normalized iso-values that range from $0.3$ to $0.8$ with distinct intervals of $0.2$, $0.3$, and $0.4$. As is shown from Figure 8b to 8f, the network takes the horizons with various iso-value annotations and the same faults to model geological structures. By visual comparison, the nearly identical predictions indicate that the modeling accuracy is not sensitive to the changes of horizon annotations within a reasonable range, which is what we expect. It is worth noting that the sharp iso-value jumps exist near the right borders of the sections in Figure 8e and 8f, although there is no data to support creating those discontinuities. The undesired features are associated with the commonly used zero-padding operation in each convolutional layer of the CNN, which maintains the spatial resolution of the hidden representations but degrades the modeling accuracy near the boundaries. To mitigate such boundary effects that potentially introduce instabilities, a straightforward way is to predict a model in a larger space and cut off the bounding regions from the prediction. Additionally, in comparison of Figure 8c to 8e, we also observe that a larger gap of the horizon annotations causes a more significant displacement of geological layers on the opposite of faults in the predicted model. Based on this observation, we recommend labeling the scattered points on each horizon with the average vertical coordinate to correctly model the stratigraphic sequences of geology. Note that the horizon annotations require to be consistently rescaled by the model size for keeping compatible with the normalized training samples.

## 5.2   Real World 2-D Case Studies

We apply the trained CNN to 2-D field data to verify its modeling performance for the structural data with geometrical patterns distinct from the training data. The input structural data are manually interpreted from the seismic images that are acquired from the Westcam dataset. This dataset is acquired in regions with closely spaced and complexly crossing faults with large slips, in which the seismic images are of low resolution due to insufficient coverage and data stacking. The ambiguous seismic reflections shown in Figure 9a are difficult to be continuously tracked across the entire seismic images, which causes the partially missing horizon data displayed by different colors. The faults shown in Figure 9b also might not be fully detected from the seismic images in the presence of data-incoherent noise and the stratigraphic features that are similar to structural discontinuities. Moreover, the structural contradictions and hard-to-reconcile features in the structural data might negatively

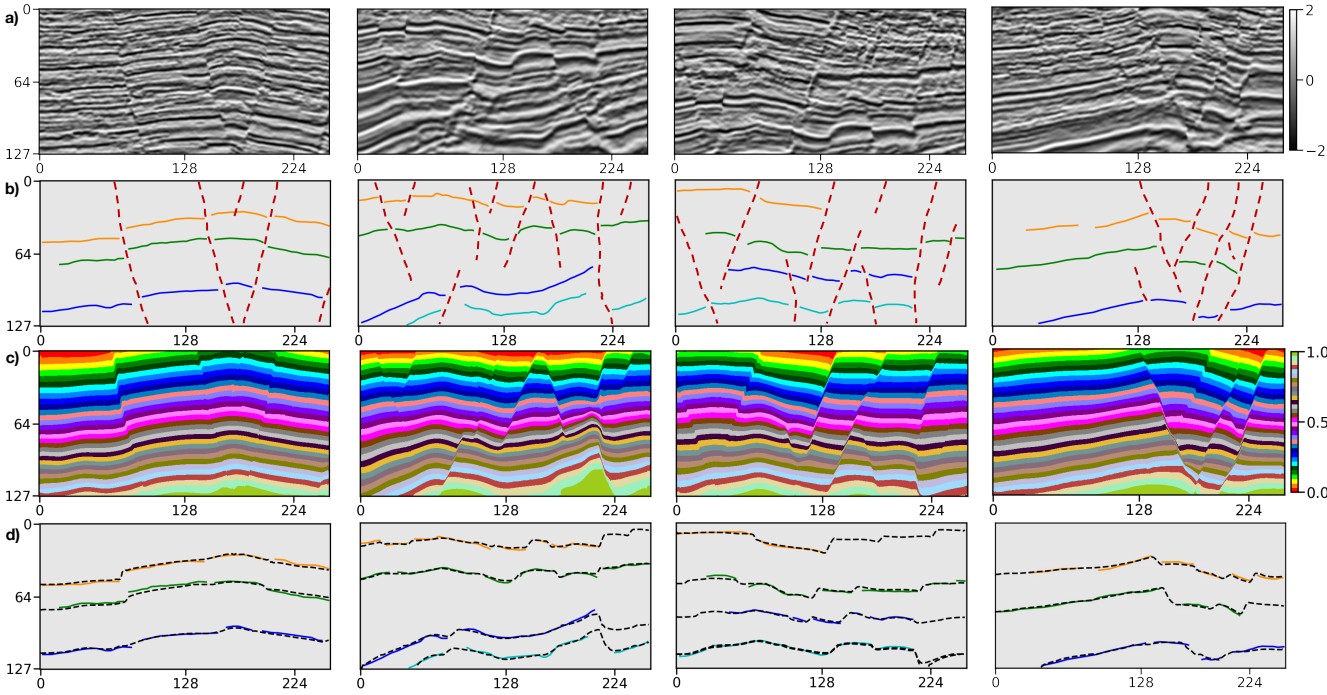

**Figure 9.** Application in a 2-D seismic field dataset. We display seismic images (a), input structural data (b) interpreted from the seismic images, predicted structural models (c), and horizon fitting results (d), respectively.

impact the modeling performance of the conventional implicit methods. Thus, there remains a challenging task to obtain a geologically reasonable model that is structurally compatible with the inputs.

As is shown in Figure 9c, our method provides geologically valid models, of which the structural discontinuities and the stratigraphic interface variations honor the input fault and horizon data. Figure 9d illustrates that the iso-lines (black lines) extracted from the modeling results consistently match the horizon interpretations (Figure 9b), which again supports our previous observation. In comparison with the scattered point-sets, a full structural model is more useful to well understand geological structures and qualify reservoir properties of continuity and morphology. Furthermore, the deep learning method with high computational efficiency can even produce real-time prediction to correct interpretation errors and improve the geological consistency of the model by taking all the structural interpretations into account.

The second 2-D field data experiment uses the dataset acquired from a geological survey and mineral exploration of the Araripe Basin in the region of the Borborema Province of Northeastern Brazil (Fabin et al., 2018). We collect the outcrop observations shown in Figure 10a and 10f from exposures of quarries on the southwestern and northern borders of the basin. There exist a series of moderate to high angle faults caused by local subsidence due to the syn-depositional dissolution of the gypsum in a large exposure of deposits of the Romualdo Formation. These syn-depositional faults control the lateral thickness variations of the stratigraphic interfaces that the field observations sample from outcrops. The field observations are integrated

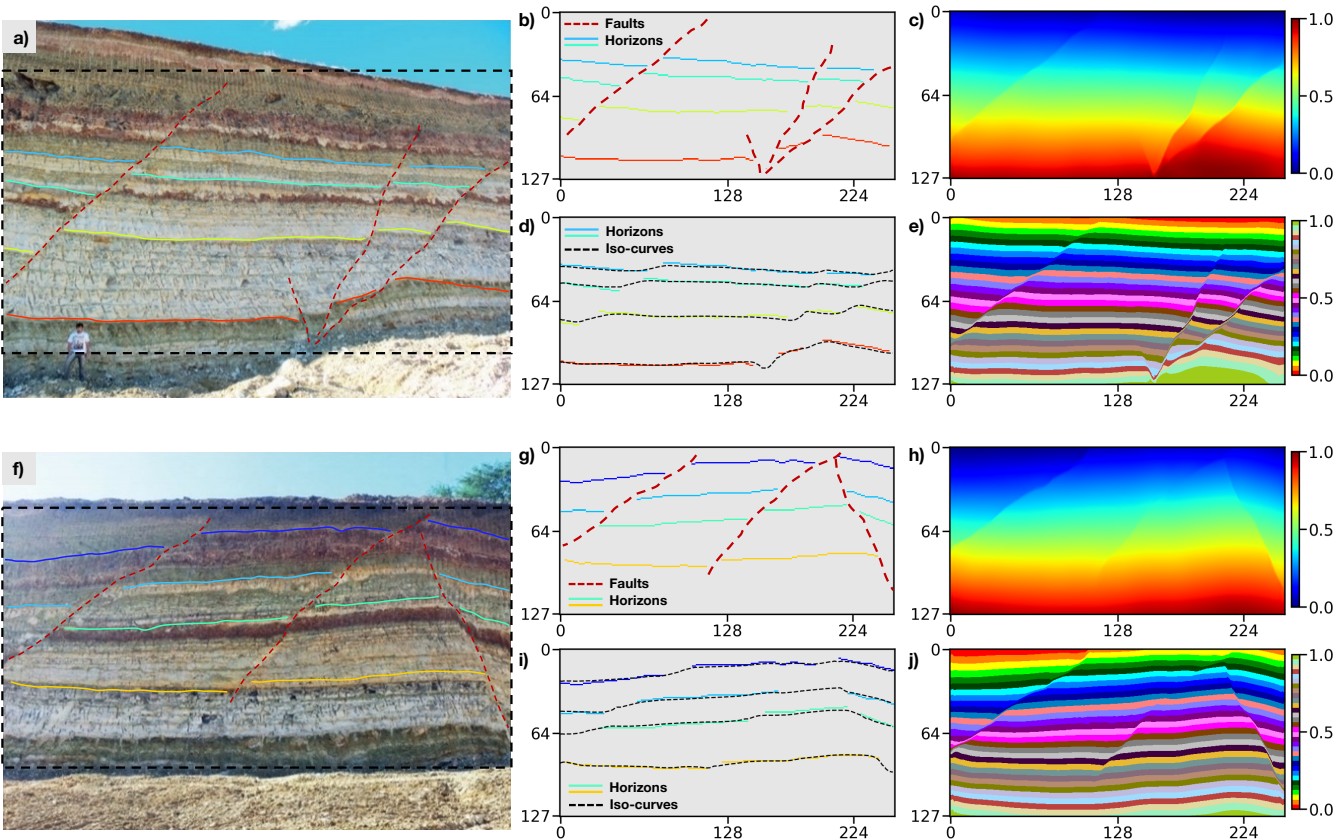

**Figure 10.** Application in an outcrop field dataset. We display two outcrop images (Fabin et al., 2018) in (a) and (f), input structural data in (b) and (g) manually interpreted from the outcrop images within the dashed boxes, predicted structural models using continuous color map in (c) and (h) and discrete color map in (e) and (j), and horizon fitting results in (d) and (i), respectively.

into the uniform sampling grids and used as inputs of our trained CNN to predict a full structural model. As is displayed in
Figure 10b and 10g, the dataset incorporates four stratigraphic interfaces and three or four faults, exhibiting a structural pattern of syn-depositional deformation that is not included in our training dataset.

The modeling results presented in Figure 10c and 10h are obtained from the trained CNN, in which all the sharp edges are structurally consistent with the faults shown in Figure 10b and 10g, respectively. The same structural models are displayed in Figure 10e and 10j using a discrete color map to indicate dislocated stratigraphic layers on the opposites of the faults. The
iso-lines extracted from the modeling results that correspond to the four distinct horizons are shown in Figure 10d and 10i, respectively. Both figures highlight the excellent fitting characteristic of the trained CNN on the input structural data. Therefore, although the CNN is trained with the automatically simulated data, it still provides a promising performance on the real-world dataset acquired at totally different surveys with complex geological structures.

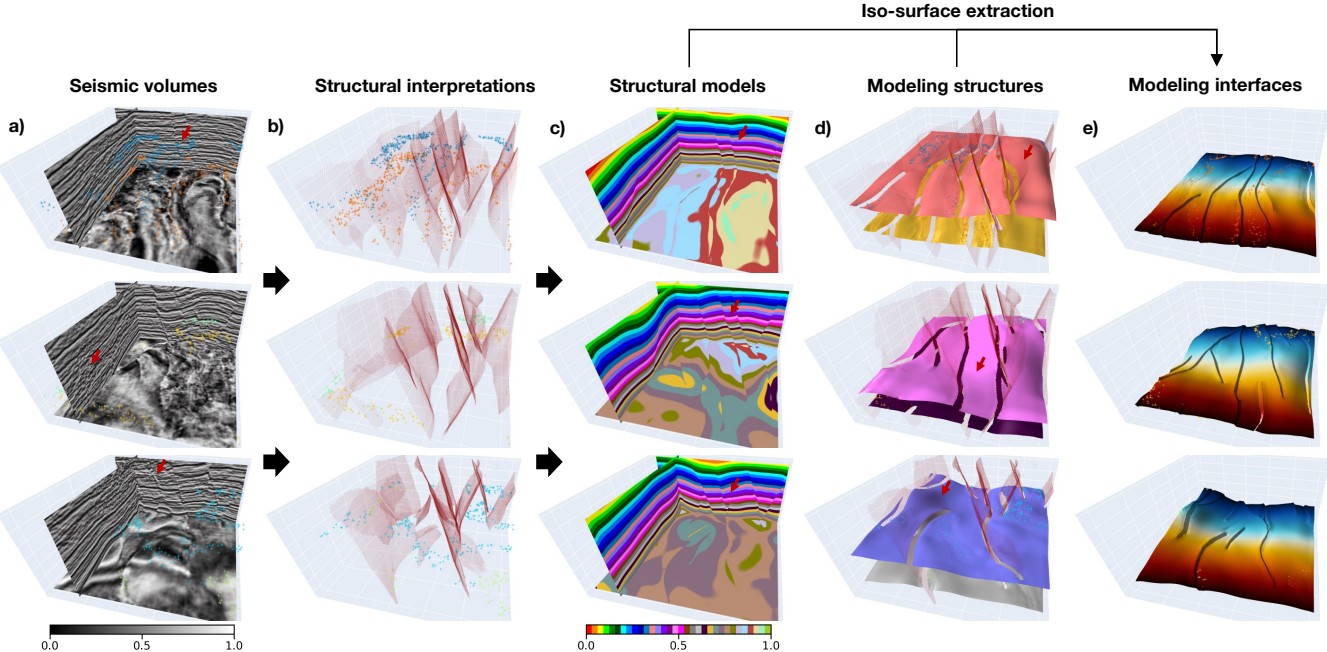

**Figure 11.** The first real-world data application. We display seismic volumes (a), input horizon and fault data (b) interpreted from the seismic volumes, modeling results (c), modeled geological interfaces extracted from the predictions and overlaid with the faults (d), and one of the recovered full horizon surfaces colored by vertical coordinates (e), respectively.

## 5.3 Real World 3-D Case Studies

Using the automatically simulated dataset, we train a 3-D modeling network with the same architecture as the 2-D CNN above to correctly capture the geometrical characteristics of 3-D geology. To validate its modeling ability, we apply the trained CNN to 3-D field data and construct a full structural model from unevenly sampled scattered points obtained from seismic interpretation. The first seismic data sampled in regions with complexly deformed structures have relatively low resolution and signal-to-noise ratio. As is shown in Figure 11a, some seismic reflections are noisy and difficult to be continuously tracked

across the entire volume of interest. The closely spaced and crossing faults further complicate the geometries of structures especially when there exists data-incoherent noise and stratigraphic features that are similar to structural discontinuities (highlighted by arrows). In addition, the scattered points heterogeneously sampled around the geological interfaces (Figure 11b) are sparse or clustered in some localized regions because of the large variations in the distances between the points.

The modeling results shown in Figure 11c demonstrate that the CNN architecture is beneficial for 3-D structural modeling

by providing a geologically valid model. We extract the full geological interfaces from the resulting scalar fields by using the iso-surface extraction method and mask the surface segments near the faults to highlight the structural gaps due to faulting in Figure 11d. Figure 11e displays a single modeled interface without masking and colored by vertical coordinates, in which there exist sharp vertical jumps across the faults. As is displayed in Figure 11d and 11e, the modeled structural discontinuities

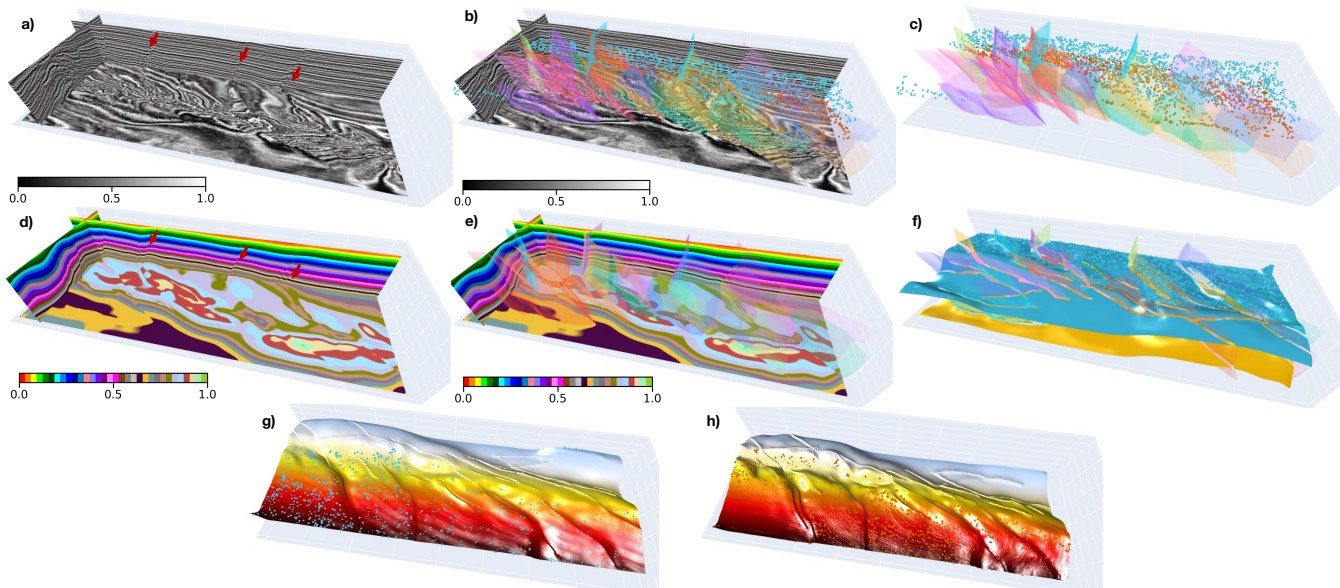

**Figure 12.** The second real-world data application. We display seismic volume (a) overlaid with structural data (b) that are interpreted from the seismic volume, input horizon and fault points (c), modeling result (d) overlaid with faults (e), modeled interfaces extracted from the prediction (f), and each of the recovered full horizon surfaces in (g) and (h), respectively. Noting that we rotate the modeled interfaces in (g) and (h) by $180°$ and color with vertical coordinates for a better visual comparison.

and interfaces can be compatible with the inputs, and the predicted models even maintain the folding structural variations
(highlighted by arrows) without global plunge information used to constrain modeling. By visual comparison, the horizon points sampled in the same geological layer are distributed around the corresponding iso-surface of the predicted model, which again demonstrates a great fitting characteristic of our network.

The second 3-D real-world case study is of a conformably folded and layered model with numerous faults that are curved and complexly intersected with each other. As is shown in Figure 12a, the available horizon data are manually interpreted on
the two stratigraphic interfaces, while the fault data are derived by using the automatic fault detection method from the seismic volume (Figure 12b and 12c). In our CNN's prediction shown in Figure 12d, the geological layers represented by iso-values with the same color accurately follow the tendency of seismic structural variations near the faults (highlighted by arrows in Figure 12a and 12d) even though we do not input any seismic data in our CNN. We also display the modeling result overlaid with the input fault data in Figure 12e, from which we can observe dislocations of geological layers on the opposites of the fault
structures (highlighted by arrows in Figure 12e). In Figure 12f, we extract the geological interfaces that correspond to the input horizons using the iso-surface extraction approach while masking the surface segments across the faults. To show more details, Figure 12g and 12h display each of the two modeled interfaces, by which we rotate $180°$ and color with vertical coordinates

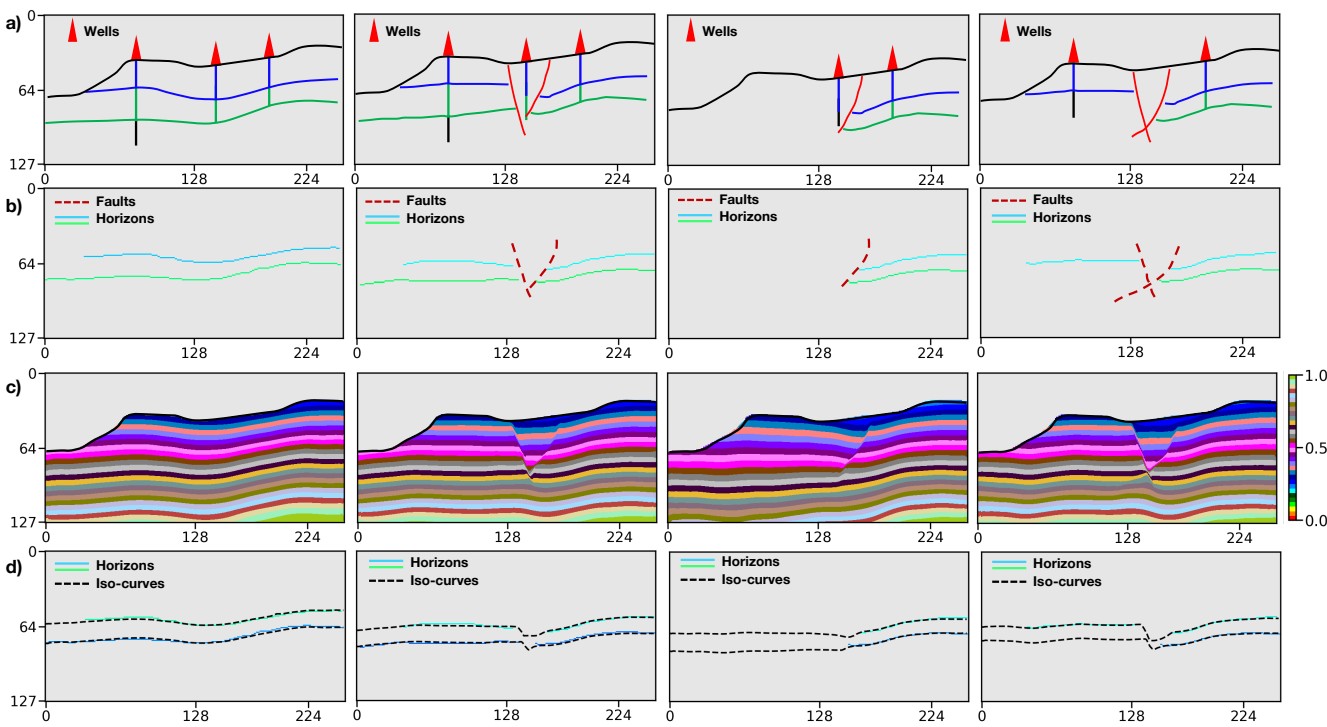

**Figure 13.** Geological uncertainty analysis. We display multiple sets of modeling elements interpreted from the vertical boreholes and the outcrop observations (a), input horizon and fault data (b), modeling results (c), and iso-curves extracted from the predictions (d), respectively.

for a better visual comparison. A great consistency between the input and recovered structures highlights the CNN's fitting characteristics on the given geological knowledge and structural constraints.

## 6  DISCUSSION

In this section, we discuss the modeling characteristic of our method and its abilities for structural uncertainty analysis, along with the current limitations. We also demonstrate a potential improvement that we will focus on in future research to incorporate extra structural constraints in the CNN-based structural modeling.

### 6.1  Structural Uncertainty Characterization

When modeling complex geological structures, the reliability of the implicit methods is heavily dependent on the quality and availability of the input structural data. However, the heterogeneously distributed structural data pose an ill-posed problem that there exist multiple plausible structural models which equally fit the inputs. Therefore, data uncertainty analysis is necessarily critical to looking for an optimal solution, especially for the noisy and hard-to-reconcile structural observations (Viard et al., 2011; Lindsay et al., 2012). Although the existing implicit methods can generate various models by perturbing the inputs to

characterize uncertainties, they might not explore a broad range of possible geological patterns and structural relationships in nature through a single model suit for stochastic simulation (Jessell et al., 2022). Working on the automating of modeling workflow, our CNN is beneficial for a flexible interpretation of aleatory and epistemic uncertainties (Pirot et al., 2022) by generating diverse modeling realizations instead of one best due to its high computational efficiency.

We adopt various combinations of modeling objects with the horizons and faults interpreted from the boreholes and the
outcrop observations shown in Figure 13a to study the uncertainties associated with the position variations of geological structures. The model with the simplest structures consists of multiple continuous and conformal geological layers shown in the first data example. The modeling situations become more complex and various when considering additional geometrical objects such as faults or unconformities to dislocate the continuous layers. Furthermore, we randomly perturb the interpreted geological interfaces to yield the variations of layer thickness because the stratigraphic boundary transition often might not
be accurately observed from vertical boreholes. As is shown in Figure 13b, the network takes a diverse set of combinations to model the possible structural geometries and relationships to demonstrate proof of concept. All the modeling results shown in Figure 13c are computed by using a desktop PC with Intel Xeon-5120 CPU (2.20GHz) and a single NVIDIA Tesla V100 GPU. Although we take a few hours in training the CNN, the average time for generating each model is approximate $0.2$ seconds using a $128 \times 256$ image size. Also, we display the horizon data overlaid with the corresponding modeled interfaces
in Figure 13d, which verifies an excellent fitting characteristic of our approach on the given structures.

## 6.2 Structural Orientation Constraint

Our CNN architecture permits the flexible incorporation of varying types of geological information by defining an appropriate loss function to measure the modeling error for every structural constraint. The input data of our method are not limited to horizons and faults and can include the structural angular observations in modeling process. In this section, we use the
structural angular information that represents local orientations of geological layers to permit geometrical relationships in the gradient of the scalar function to be considered. The loss function of orientation constraint is aimed to measure the angle errors between the directional derivatives of the predicted model and the orientation observations using cosine similarity. We adopt the second-order accurate central differences method (Fornberg, 1988) using the Taylor series approximation to estimate the local orientation at each interior point of the given structural model $\mathbf{z}$. The cosine similarity at every single point between the
orientations of the predicted model and the normal vector $\overrightarrow{n}$ can be represented as follows,

$$f_{cs}^{pred}(p) = \frac{\overrightarrow{n} \cdot \nabla \mathbf{z}(p)}{\| \overrightarrow{n} \| \| \nabla \mathbf{z}(p) \|}. \tag{5}$$

This is also used to compute the cosine similarity between the orientations in the reference model $f_{cs}^{obs}(p)$ and the normal vector $\mathbf{n}$. Therefore, the loss function that measures the structural angle errors between the two models being compared can be formulated as follows,

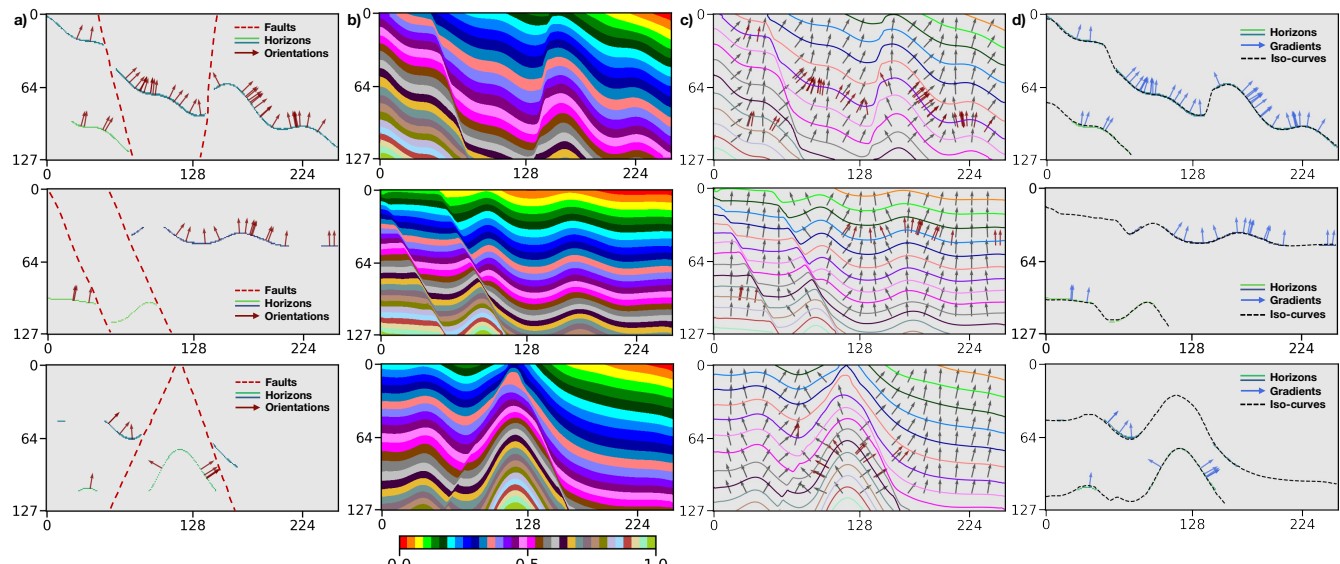

**Figure 14.** The faults and incomplete horizons, together with the structural orientations, are used as inputs (a) into the network to predict structural models (b). The sets of regularly spaced iso-lines and local orientations are obtained from the modeling results (c) and compared with the given normal vectors (red arrows). The modeled stratigraphic interfaces and their normal vectors well match the two distinct input horizons (d).

$$\mathcal{L}_{\text{normal}}(\mathbf{p}) = \frac{1}{N} \sum_{p \in \mathbf{p}} |f_{\text{cs}}^{\text{pred}}(p) - f_{\text{cs}}^{\text{obs}}(p)|, \tag{6}$$

in which $N$ represents the total number of the points within a patch $\mathbf{p}$. Therefore, the total loss function is defined by combining the various types of geological constraints as,

$$\mathcal{L}_{\text{sum}}(\mathbf{p}) = \frac{1}{K} \sum_{i=1}^{K} (\lambda \mathcal{L}_{\text{mae}}(\mathbf{p}_i) + \mathcal{L}_{\text{ms-ssim}}(\mathbf{p}_i) + \beta \mathcal{L}_{\text{normal}}(\mathbf{p}_i)), \tag{7}$$

where $\beta$ is used to balance the relative significance of the orientation loss, and $K$ represents the number of the patches cropped from the model. $\beta$ and $\lambda$ are empirically set to $1.00$ and $1.25$, respectively, according to many prior numerical tests. As is displayed in Figure 14a, the faults and the incomplete horizons, together with the structural orientations sparsely distributed on the horizons, are used as inputs to the network to model geological structures. The modeling results shown in Figure 14b exhibit the remarkable performance of our method when using various types of geological data inputs. We extract uniformly spaced geological interfaces and their local orientations from every modeling result and compare them with the input horizons and normal vectors (red arrows) in Figure 14c, which presents a great consistency between the predicted and input structures. The interpolated structures by using the CNN maintain the large localized geometrical variations even though there is no

global geological information to constrain the modeling process. In addition, the iso-lines and normal vectors (blue arrows) extracted along the stratigraphic interfaces of the predictions well match the two distinct input horizons (Figure 14d), which again highlights the CNN's fitting characteristic on the given structural constraints.

## 6.3  Current Limitations and Improvements

The CNN trained by using the synthetic dataset presents excellent modeling capacities in real-world cause studies to represent complicated geological structures that are distinct from the simulated models. Instead of imposing any explicit mathematical constraints in the conventional implicit methods, our CNN-based structural modeling is implemented by the recursive spatial convolutions with trainable kernel parameters and the loss function related to various geological constraints. The spatial convolutions in the CNN can be viewed as the implicit interpolants used in the traditional approaches, and the only difference is that the parameterization of their kernel functions is optimized through training. As structural modeling is dependent on the analysis of the spatial relations of the observed structures to interpolate new geologically valid structures elsewhere, acquiring representative example data is essential for training the CNN to achieve its reliable generalization performance. Therefore, we adopt an automatic workflow to generate numerous models with realistic structures and simulate partially missing horizons in building the training dataset. It is a significant reason why our network could be applied to the real-world datasets acquired in different geological surveys with distinct structural patterns.

Another improvement of our approach is attributed to the use of a hybrid loss function based on element-wise accuracy and structural similarity when updating the CNN's parameters. To demonstrate the improved modeling performance, we implement a quantitative analysis of our CNN trained with the different loss functions using the multiple quality metrics. The averages of these metrics on the validation dataset are tabulated in Table 2. The CNN trained with the hybrid loss function of MS-SSIM and MAE (denoted by MS-SSIM&L1) can outperform the others in Table 2 on all the quality metrics even including the quality metrics which we use as cost function to train the network. This loss function is attributed to a better reconstruction of fault-related features in the resultant model by assigning high weights to regional structural contrasts. Also, reliable identification of faults is useful to constrain the lateral occurrence of stratigraphic interfaces across structural discontinuities.

Although working well to recover faulted and folded structures, the proposed method might not properly represent other geological structures that are not considered in the training dataset, such as unconformities and igneous intrusions. The trained network also might not correctly construct low dip-angle thrust faults in predicted models because we still do not include this type of fault in the used data generator. Despite the current limitations, the proposed CNN architecture still shows promising potential to compute a geological valid and structurally compatible model honoring the observed structures. Considering the used training samples are still not sufficiently diverse to support modeling complex and unseen geological settings, future works will focus on expanding the training dataset to a broader range of structural geometries and relationships associated with these settings. For example, we can further augment simulation workflow by adding more complex and various features in the structural models, or adopting a recently developed 3-D geological modeling dataset (Jessell et al., 2022) where dykes, plugs, and unconformities are incorporated.

## 7    Conclusions

A CNN-based deep learning method has been used to represent geological structures over the entire volume of interest from typically sparse and hard-to-reconcile structural interpretations. The network is composed of encoder and decoder branches and supplemented with lightweight depth-wise separable convolution and channel-wise attention to find an optimal trade-off between modeling accuracy and computational efficiency. The developed CNN architecture leverages the low-rank nature of the sparse and heterogeneously sampled structural data to adaptively suppress uninformative features by using a linear bottleneck and inverted residual structure in each of the encoded convolutional layers. Our approach is beneficial for the flexible incorporation of empirical geological knowledge constraints in a supervised learning framework using numerous and realistic structural models that are generated from an automatic data simulation workflow. This also provides an impressive characteristic to flexibly integrate multiple types of structural constraints into the modeling by using an appropriate loss function, exhibiting a promising perspective for further improving geological modeling. We verify the effectiveness of the proposed approach by using the case studies acquired in distinct geological surveys, including synthetic examples created by the same workflow for acquiring the training dataset, the randomly created modeling objects without any ground truth of geology, and the structural interpretations obtained from the seismic images. In both synthetic data and real-world data applications, we verify its modeling capacities in representing complex structures with a model geologically reasonable and structurally compatible with the inputs.

*Code and data availability.*  The synthetic structural models, used for training and validating our network, are uploaded to Zenodo and are freely available through the DOI link https://doi.org/10.5281/zenodo.6480165. The source codes for the neural network developed in Pytorch are uploaded to Zenodo and provided at the DOI link https://doi.org/10.5281/zenodo.6684269.

## Appendix A:  Regression metric functions

To verify the modeling performance of our CNN, we quantitatively measure the differences between the ground truth structural models and predictions by using various regression metrics including SSIM (Structural Similarity), MSE (Mean Square Error), MAE (Mean Absolute Error), EVS (Explained Variance Score), MSLE (Mean Square Logarithm Error), MDAE (Median Absolute Error), and R2S (R Square Score, also called the Coefficient of Determination) in the validation dataset. MSLE measures the prediction performance that corresponds to the expected value of the squared logarithmic error, which is formulated as,

$$f_{\mathrm{MSLE}}(\mathbf{y}, \hat{\mathbf{y}}) = \frac{1}{N} \sum_{i=1}^{N} (\log_e(1 + \mathbf{y}_i) - \log_e(1 + \hat{\mathbf{y}}_i))^2, \tag{A1}$$

where $\mathbf{y}$ and $\hat{\mathbf{y}}$ are the structural models being measured, respectively, and $N$ represents the total number of points within the model. MDAE is computed by using the median of all absolute differences and thus can be robust to outliers,

$$f_{\text{MDAE}}(\mathbf{y}, \hat{\mathbf{y}}) = \frac{1}{N} \sum_{i=1}^{N} \text{median}(\mathbf{y}_i - \hat{\mathbf{y}}_i). \tag{A2}$$

EVS is used to measure the proportion of the variability in the solutions of a machine learning method, and its score value
ranges from zero to one. Higher EVS typically indicates a stronger strength of association between regression targets and predictions and thus represents better network performance. It can be formulated as follows,

$$f_{\text{EVS}}(\mathbf{p}) = \frac{1}{M} \sum_{p \in \mathbf{p}} (1 - \frac{\text{variance}(\mathbf{y} - \hat{\mathbf{y}})}{\text{variance}(\mathbf{y})}), \tag{A3}$$

where $\mathbf{p}$ represents the patch cropped from the same spatial location from the structural models being measured, and $M$ is the number of the cropped patches. R2S offers a measurement of how well the predictions of the network are based on the
570 proportion of total variations. R2S can be written as follows,

$$f_{\text{R2S}}(\mathbf{p}) = \frac{1}{M} \sum_{p \in \mathbf{p}} (1 - \frac{\sum_{i=1}^{N} (\mathbf{y} - \hat{\mathbf{y}})^2}{\sum_{i=1}^{N} (\mathbf{y} - \overline{\mathbf{y}})^2}), \tag{A4}$$

in which $N$ represents the total number of points within the cropped patch $\mathbf{p}$. R2S is similar to EVS, with the notable improvement that it accounts for systematic offsets in the solutions. In addition, EVS and R2S can be more robust and informative than MAE and MSE in regression analysis evaluation as the former can be represented as percentage errors.

*Author contributions.* XW initiated the idea of this study and advised the research on it. ZB conducted the research and implemented the 2-D and 3-D CNN-based structural modeling algorithms. XW conducted numerical structural simulations to provide synthetic structural models for training. ZB prepared the training datasets from the simulated structural models and carried out the experiments for both synthetic and real-world case studies. ZL, DC, and XY helped design the experiments and advised on result analysis from a geological perspective. ZB and XW prepared the manuscript with contributions from all co-authors.

*Competing interests.* The authors declare that they have no conflict of interest.

*Acknowledgements.* This research is financially supported by the National Science Foundation of China under grant no. 42050104, 41974121 and Research Institute of Petroleum Exploration & Development-NorthWest (NWGI), PetroChina.

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
