# Peer review of "DeepISMNet: Three-Dimensional Implicit Structural Modeling with Convolutional Neural Network"

_Geoscientific Model Development, 2022_

## Referee Comment (RC1)

Review of

**DeepISMNet: Three-Dimensional Implicit Structural Modeling with Convolutional Neural Network**

Zhengfa Bi1, Xinming Wu1, Zhaoliang Li2, Dekuan Chang3, and Xueshan Yong

https://doi.org/10.5194/gmd-2022-117

The manuscript is well-written, but needs additional editing to correct some grammatical errors.

Appropriate figures, but Fig 11 and 12 need to be enlarged. Annotation would help to refer the reader to the specific examples mentioned in text (I elaborate in the comments below)

I attempted to test the github code on a Windows operating system and had a few issues with conflicting dependencies with installed modules. In particular, I received open MPI errors:
```
"OMP: Error #15: Initializing libiomp5md.dll, but found
libiomp5md.dll already initialized."
```
In attempting to fix this, a number of other dependencies now no longer work.

Most figures appear before being cited in text.

**Introduction**

Line 26 Calcagno et al 2008 is a more appropriate reference than Guillen et al 2008. Guillen focusses on geophysics, not so much on structural modelling.

Line 28 "interpreting" should be "interpolating"?

Line 33 Please elaborate on what you mean by "empirical rules" – is this the tectonic history or topology (i.e. stratigraphy; fault-fault and fault-stratigraphy relationships)? See Calcagno et al 2008 for one example)

Line 33 "file observation" is "field observation"?

Line 37 – "or regional available" - this part of the sentence doesn't make sense

Line 47 "DIS" should be "DSI"

Lines 56 – 69
The authors make some valid points. Also worth mentioning is implicit methods suffer with dense and clustered data points, which can be addressed through downsampling. The problem is that downsampling brings its own issues, such as how or whether to aggregate data points and what appropriate algorithms from which to produce representative inputs to implicit methods. I suppose this is what you describe Lines 128-130?
Line 61-62 Also look at Grose et al 2018; 2021 (10.1029/2017JB015177; 10.5194/GMD-14-3915-2021) for implicit methods that incorporate greater number of relevant structural data as constraints.

Line 70 -73
The authors need to explain how "past experiences" are relevant to structural modelling. What is 'example data'? Observations, or other models (in which case, not data). Providing examples would help here.

Line 74 "without defining a physical process". Implicit modelling doesn't require a description of a process, but geometrical and topological constraints. The resulting model geometries can be interpreted to represent some process, but that is not quite what is written here. Process models are an entirely different class to structural ones, at least in the context described earlier in the document.

Line 78 citation "Ioffe and Normalization" is Normalization really the name?

Line 88 You imply that GNNs being not repeatable is a problem... I assume this means that your method is? Perhaps state how your NN architecture is able to ensure repeatability (presumably because the parameters are not randomly initialised?)

Line 99 How is the scalar field produced? Automated model building (how?) Masking why? Many questions about where the rendering happens

**Method**

Skip connections are an important feature in your NN implementation. You need to explain how they work and why they are necessary so that the reader understands how they allow what appears to be multiscale modelling.

Line 168 What is "MAdds" referring to?

* Line 180. Inverse-distance weighting schemes are well-used in spatial stats, but just because it's consistent with traditional methods, doesn't mean it's appropriate. For example, what about in geological scenarios where there is a large shear zone separates one geological terrane from another? The data on each side of the shear zone may be spatially close, but may be quite distant contextually. So, the assumption of weighting on spatial proximity goes against some of your criticisms of existing methods in the introduction. It's a known drawback of other implicit geomodelling schemes. Justifying this assumption is important, and please provide supporting references. In fairness, this is quite a difficult

problem that I don't believe you're attempting to solve directly with this contribution, Nonetheless, it's worthwhile discussing as a potential limitation.

Line 190 provide references for which methods use MSE/MAE.
Line 204 provide references for SSIM
Line 210 the term G_sigma_g is missing from eq. 2? As is sigma_g
Lines 215 I see that sigma_g is a 'superparameter'. It may help to define this as a 'hyperparameter' to fit with ML semantic conventions, unless there is a specific reason you're using 'super'. If so please explain why 'super' and not 'hyper'.
Line 231 'artifacts' are old and interest physical objects from antiquity, 'artefacts' are unintended side-effects from numerical processes.

Line 231 – what is 'it' in the phrase "such as MAE to the MS-SSIM as *it* is only related to the values on a single point" the artefacts? Or the MAE... ?

**Data preparation**

Given how important the training data is for your procedure, you need to have at least a two or three sentences explaining the method of Wu et al 2020. Wu et al 2020 generates a training set using a CNN for your CNN... Both methods produce a scalar field from which to render geology – so it would be helpful to make clear how your approach differs.

Line 262. This is a question commonly asked, but needs to be addressed. How do you know 600 models is enough given the uncertainties of a result that you admit cannot be easily validated (L253)?

**Implementation**

Section 4.1 Training and validation

I understand the need for normalisation, but it's not clear how this is achieved on the entire model. Structural data, at least in most implicit schemes consists of contact observations (X,Y,Z coordinates, or U,V,W and a 'lithology' label with topology, usually a stratigraphic direction or polarity) and orientation data (X,Y,Z, type [contact orientation; fault orientiation; fold hinge; etc] a vector; usually expressed as dip direction/dip and the lithology it represents). How do you simultaneously normalise coordinates and vectors? What is method for normalisation e.g. min-max?

Lines 328-329 You do explain some of these metrics elsewhere in the manuscript, but not all e.g. R2S. You may think to supply these explanations as supplementary material.

Line 339 – 346 This paragraph belongs in the discussion.

Line 348-350. The context of this problem isn't clear. Annotate with what, and do other applications do this, but poorly, or not do it at all? What do you mean horizon values – it would seem that it is the iso-value from the scalar field (or fields, especially if considering faults..). Provide some examples with relevant citations.

**Application**

Section 5.1 Provide a sentence describing the geological scenario and location of study area #1.

2d case studies are simple and not a great test as the horizons are laterally contiuous being essentially flat. The fault displacements seem to be honoured, which is good. But existing model packages are very effective at producing models with similar simple geometry and topology.

Line 401 – how do you define complex geological settings here? Polydeformed beds? Domains with vastly different tectonic histories? Multiple magmatic intrusion and extrusion events? Overturned bedding?

Figs 11 and 12 have the necessary content, but are too small. Enlarge each panel in the figure, and use more space in the manuscript. Annotations would also help the reader, such as pointing out where the complex parts of the model are (see previous comment), where "the seismic reflections are partially ambiguous and difficult to be continuously tracked" and "results shown in Figure 11c demonstrate that our CNN architecture is beneficial for 3-D structural modeling by predicting a geologically valid model".

**Discussion**

Section 6.1 and Fig 13. I like the uncertainty analysis. One thing worth pointing out is that you deal with both aleatory uncertainty (uncertainties resulting from measurement error) and epistemic uncertainty (relating to missing knowledge or data) citing Pirot et al 2022 in GMD. You remove/add drill holes in your fig 13, which is a nice test, so it is worth highlight that point here, or at the very least in the figure caption.

Lines 484 – 487. This is good to acknowledge as modelling these types of structures realistically is challenging, however you can elaborate on why your method is unable to replicate them. I would think the type of geological object is arbitrary given the framework you developed. Is it not possible because the synthetic model generation of Wu et al 2020 cannot recreate unconformities and intrusions? A possible solution maybe to expand the training set to a wider range of objects? Another source could be the Noddyverse (Jessell et al .2022) where dykes, plugs and unconformities are represented. Perhaps you can comment in whether this would work of not. It could be that because you're using a scalar field in the tradition of Lajuanie et al 1997, in which intrusions and unconformities (as distinct from onlap relationships) have not yet been solved (at least to my knowledge).

---

## Author Comment (AC1)

Responses to comments from reviewers

Dear reviewer,

We sincerely appreciate all your careful reviewing so that we could get the reviewed manuscript promptly. We appreciate all your valuable comments and suggestions, which help a lot to improve our manuscript. We have corrected the figures and spelling mistakes and made modifications in the manuscripts according to your comments. Below we are trying to responses all your comments, suggestions, and questions. Let's discuss more if some of our explanations in the responses are not clear to you. The related modifications are not shown in the responses but are all marked in the manuscript revision history.

Thanks!

● **Introduction:**
1) Line 26: Calcagno et al 2008 is a more appropriate reference than Guillen et al 2008. Guillen focusses on geophysics, not so much on structural modelling.
   Thanks. We have modified reference citation according to your suggestion.

2) Line 28: "interpreting" should be "interpolating"?
   Thanks for your suggestion. We have modified the related words.

3) Line 33: Please elaborate on what you mean by "empirical rules" – is this the tectonic history or topology (i.e. stratigraphy; fault-fault and fault-stratigraphy relationships)? See Calcagno et al 2008 for one example).
   Thanks for your comments. We have added the corresponding texts to demonstrate "empirical rules" for improving readers' understanding to the content. The empirical knowledge can be mentally inferred from the structural data to constrain the possible geometrical relationships among the interfaces of geological formations and drive the modeling behavior of the implicit methods. The input structural data of the implicit method typically consists of various types of modeling objects, such as spatial points, vectors, polylines, and surfaces, interpreted by the geologists and geophysicists from field observations. The implicit method benefits from incorporating available geological information into the resultant model by integrating the observed data and the empirical knowledge, providing an effective alternative to reproduce the geometry of the subsurface from a global respective.

4) Line 33: "file observation" is "field observation"?
   Thanks for your comment. Corrected.

5) Line 37: "or regional available" - this part of the sentence doesn't make sense.

Thanks for your suggestion. We have deleted "or regional available" in the correspond content of our manuscript.

6) Line 47: "DIS" should be "DSI".
   Thanks, Corrected.

7) Lines 56 to 69: The authors make some valid points. Also worth mentioning is implicit methods suffer with dense and clustered data points, which can be addressed through downsampling. The problem is that downsampling brings its own issues, such as how or whether to aggregate data points and what appropriate algorithms from which to produce representative inputs to implicit methods. I suppose this is what you describe Lines 128-130?
   Thanks for your comment. We have added the related discussions in section "Introduction" of the manuscript to supplement the potential challenges of "clustered data points" in the existing implicit approaches based on your suggestions. On the one hand, structural interpolation fully guided by mathematical equations might not always produce a geologically valid model given sparse data in some complex geological circumstances. On the other hand, the implicit methods also suffer from the unevenly distributed data, such as dense or clustered points, in which how to appropriately aggregate the structural information and produce representative inputs is still an active area of research.

8) Line 61 to 62: Also look at Grose et al 2018; 2021 (10.1029/2017JB015177; 10.5194/GMD-143915-2021) for implicit methods that incorporate greater number of relevant structural data as constraints.
   Thanks for your comment. We have added the reference citations associated with the implicit methods that incorporate different relevant structural constraints in our manuscript.

9) Line 70 to 73: The authors need to explain how "past experiences" are relevant to structural modelling. What is 'example data'? Observations, or other models (in which case, not data). Providing examples would help here.
   Thanks for your comment. We have added the related texts to demonstrate how 'example data' are relevant to structural modeling in the manuscript. In the structural modeling, the 'example data' are considered as numerous geological models with full structural annotations, which are used to create a sufficiently large dataset for supervising the optimization of the CNN model parameters. Deep learning is a type of data-driven and statistical approach that estimate an implicit function that maps inputs to outputs based on past experiences or example data by optimizing a given quality criteria or loss function. To build example or training dataset, we first use an automatic data simulation workflow to generate numerous geological models with realistic faulted and folded structures that are not limited to a specific pattern by randomly choosing simulating parameters within reasonable ranges. Then we randomly some mask parts of the synthetic models for

generating the sparsely and unevenly distributed horizon data, which together with the fault data are used as inputs of our CNN-based modeling process.

10) Line 74: "without defining a physical process". Implicit modelling doesn't require a description of a process, but geometrical and topological constraints. The resulting model geometries can be interpreted to represent some process, but that is not quite what is written here. Process models are an entirely different class to structural ones, at least in the context described earlier in the document.
Thanks for your comment. We have deleted the phrase that confuses the readers and corrected the related sentence as, "By contrast with the traditional approaches, deep learning is beneficial for making a prediction without solving a sophisticated linear system of equations with prescribed mathematical constraints at cost of expensive computation".

11) Line 78: citation "Ioffe and Normalization" is Normalization really the name.
Thanks. We have modified the corresponding reference citation.

12) Line 88: You imply that GNNs being not repeatable is a problem… I assume this means that your method is? Perhaps state how your NN architecture is able to ensure repeatability (presumably because the parameters are not randomly initialised?)
Thanks for your comment. The CNN can produce a structural scalar field as an implicit representation of all the geological structures from various types of data based on the knowledge learned from a large training dataset. In comparison to GNN, the solutions of our approach can be reproducible as it is not necessary to randomly initialize the parameters of the network at each modeling process. Once trained, we can observe that the CNN can efficiently provide a geologically reasonable structural model in the synthetic and real-world data applications, showing promising potential for further leveraging deep learning to improve many geological applications.

13) Line 99: How is the scalar field produced? Automated model building (how?) Masking why? Many questions about where the rendering happens.
Thanks for your comment. We have modified the related texts to improve the readers' understanding to the whole workflow of our CNN-based method. Our network can produce a structural scalar field as an implicit representation of all the structures from various types of structural data such as horizons that encode the stratigraphic sequence of the sampled interfaces, and faults in the presence of the geological boundaries. To achieve this, we are first required to prepare a training dataset. As discussed in the manuscript, we use an automatic data simulation workflow to generate numerous geological models with realistic faulted and folded structures that are not limited to a specific pattern by randomly choosing simulating parameters within reasonable ranges. Then we utilize a hybrid loss function that combines element-wisely fitting on the known values and multi-scale structural

similarity over the local sliding windows to train the network. Once trained, the CNN can efficiently provide a geologically reasonable structural model from the heterogeneously sampled data based on the knowledge learned from the training dataset and implicitly embedded in the CNN model parameters.

- **Method:**
1) Skip connections are an important feature in your NN implementation. You need to explain how they work and why they are necessary so that the reader understands how they allow what appears to be multiscale modelling.

   Thanks for your comment. We have modified the related texts in the subsection "Network Architecture" of our manuscript. Our developed CNN architecture uses a U-shaped framework with the encoder and decoder branches linked by using skip connections. In such a framework, the features are first downsampled at different spatial resolutions in the encoder and then recombined with their upsampled counterparts through skip connections in the decoder. In general, the localized components of the inputs are typically extracted at an early stage of the CNN, while the relatively high-level and global features are obtained when the receptive fields are increasingly large in deep convolutional layers. Therefore, as the hidden representations with different spatial resolutions have much distinctive geometrical information, systematically aggregating the multi-scale features via skip connections can provide hierarchical constraints for a reliable and stable structural field prediction. Furthermore, the low-level features computed from the shallow layers can better follow the input structures than the deep high-level features because the structural information might be gradually missing in recursive feature compressions at the downsampled spatial resolutions. The use of skip connections is helpful to enhance the low-level features and produce a model structurally consistent with the inputs.

2) Line 168: What is "MAdds" referring to?

   Thanks. "MAdds" refers to the number of multiply–accumulate operations. It is typically used to measure the computational complexity of the method.

3) * Line 180: Inverse-distance weighting schemes are well-used in spatial stats, but just because it's consistent with traditional methods, doesn't mean it's appropriate. For example, what about in geological scenarios where there is a large shear zone separates one geological terrane from another? The data on each side of the shear zone may be spatially close, but may be quite distant contextually. So, the assumption of weighting on spatial proximity goes against some of your criticisms of existing methods in the introduction. It's a known drawback of other implicit geomodelling schemes. Justifying this assumption is important, and please provide supporting references. In fairness, this is quite a difficult problem that I don't believe you're attempting to solve directly with this contribution, Nonetheless, it's worthwhile discussing as a potential limitation.

Thanks for your comment. We have modified the corresponding paragraph in the subsection "Network Architecture" of our manuscript. Our CNN is designed to progressively complete structural features layer by layer through sequential non-linear convolutional filters that are conditioned on the previous convolutions. The valid convolutional responses only exist near the input structures in the starting layer of the network. To spread geological structures elsewhere, each convolutional layer collects information from the previous layer outputs within an increasingly expanding receptive field by recursively stacking convolutions and down-sampling the input hidden features. Therefore, at the bottom layers of the CNN, the structural information in the inputs can be used to constrain the modeling process over the entire volume from a global receptive region of view. This characteristic allows the network to correctly understand the relations of the spatially distant but contextually close features. Although weighting on spatial proximity is typically used in many traditional structural interpolation methods, the nearby features are not necessarily more significant than distinct ones for making geological-related predictions. For example, when the stratigraphic terranes are located on the opposite of a large shear zone or other discontinuous structures, the correlations of distinct data points computed from a global review can be helpful to capture a more accurate structural pattern.

4) Line 190: provide references for which methods use MSE/MAE.
Thanks for your suggestion. We have added the associated reference citations in the manuscript.

5) Line 204: provide references for SSIM.
Thanks for your comment. We have added the corresponding reference paper in the manuscript.

6) Line 210: the term G_sigma_g is missing from eq. 2? As is sigma_g.
Thanks for your suggestion. In the definition of Structural Similarity (SSIM), variances and covariance are computed by using a Gaussian filter with standard deviation $\sigma_g$. Approximately, $\mu_x$ and $\sigma_x$ can be viewed as estimates of the stratigraphic sequences and structural variations in a local patch of the structural model, respectively, and $\sigma_{xy}$ measures the tendency of the two patches in models being compared to vary together, thus an indication of structural similarity. However, in the conventional SSIM, the standard deviation $\sigma_g$ of the Gaussian filter requires to be defined before training. However, the choice of $\sigma_g$ might influence the prediction accuracy because the CNN trained by using a large standard deviation might overly emphasize the local variations and generate spurious features while blurring sharp structural boundaries for a small standard deviation. Instead of fine-tuning the parameter, we use Multi-scale Structural Similarity (MS-SSIM) to compute a pyramid of patches with different scale levels defined by various standard deviations of the Gaussian filter.

7) Lines 215: I see that sigma_g is a 'superparameter'. It may help to define this as a 'hyperparameter' to fit with ML semantic conventions, unless there is a specific reason you're using 'super'. If so please explain why 'super' and not 'hyper'.

Thanks. Corrected. We have replaced "super-parameter" with "hyper-parameter" in the related sentence of the manuscript.

8) Line 231: 'artifacts' are old and interest physical objects from antiquity, 'artefacts' are unintended side-effects from numerical processes.

Thanks. Corrected.

9) Line 231: what is 'it' in the phrase "such as MAE to the MS-SSIM as *it* is only related to the values on a single point" the artefacts? Or the MAE… ?

Thanks for your comment. We have modified the related phrases in this paragraph to improve the reader's understanding to why we use a joint loss function that combines the MAE with the MS-SSIM for training the network. Although the MS-SSIM can highlight structural variations focusing on a neighborhood of grid point as large as the given Gaussian filter, it might produce artefacts in predictions because its derivatives cannot be correctly estimated near the boundary regions of the patch. This can be alleviated by supplying an element-wise criterion that is computed on a single point of the two patches being compared in the loss function. In addition, using MS-SSIM alone might not be sensitive to uniform biases, which might cause unexpected changes in stratigraphic sequences or shifts of geological interfaces in modeling results.

● **Data preparation:**

1) Given how important the training data is for your procedure, you need to have at least a two or three sentences explaining the method of Wu et al 2020. Wu et al 2020 generates a training set using a CNN for your CNN… Both methods produce a scalar field from which to render geology – so it would be helpful to make clear how your approach differs.

Thanks for your constructive comment. We have added the related discussions in the subsection "Automatic Data Generator" in our manuscript. As far as I'm concerned, one of the most significant steps of applying the supervised learning method is the preparation of many example data and especially the corresponding labels for training the network. In the structural modeling problem, the training dataset should incorporate structurally various geological models to allow the CNN to learn representative knowledge and achieve its reliable generalization in real-world applications. However, it is hardly possible to manually label all geological structures in a field survey as the ground truth of the subsurface is inaccessible. To overcome this problem, we adopt an automatic workflow to simulate geological structural models with some typical folding and faulting features that are controlled by a set of random parameters. In this workflow, we first create a flat layered model with horizontally constant and vertically monotonically increasing values as an initial model, and then sequentially add folding, dipping, and faulting structures to

further complicate the geometrical features of this model. We simulate the folding and dipping structures by vertically shearing the initial flat model using a combination of linear and Gaussian-like shift fields, while creating faulting structures through volumetric vector fields that are defined around the fault surfaces. By randomly choosing the parameters within the reasonable ranges, we can simulate numerous geological models with diverse and realistic features not limited to a specific structural pattern to enrich the training dataset. Based on the models with known features, we can simultaneously obtain the corresponding ground truth of fault data with ones on faults and zeros elsewhere. The generated models can be viewed as a structural scalar function because their iso-surfaces represent the corresponding stratigraphic interfaces while the local value jumps indicate the structural discontinuities. This is important for our next step of constructing an example dataset to train our CNN for geological structural modeling.

2) Line 262: This is a question commonly asked, but needs to be addressed. How do you know 600 models is enough given the uncertainties of a result that you admit cannot be easily validated (L253)?

Thanks for your comment. Although the CNN model is trained by using only 600 synthetic structural models, it still works pretty well to model the geological structures from the sparse and unevenly distributed structural data that are recorded at totally different field surveys. We cannot make sure the current number of the training samples is the best one, but further tuning might be very time-consuming and hardly obtain further improvements. Considering the used training dataset is still not sufficiently large to train a 3-D deep network, future works will focus on further complicating the data simulation workflow by adding more complex and representative geological patterns in the synthetic structural models.

- **Implementation:**

1) Section 4.1: Training and validation. I understand the need for normalization, but it's not clear how this is achieved on the entire model. Structural data, at least in most implicit schemes consists of contact observations (X,Y,Z coordinates, or U,V,W and a 'lithology' label with topology, usually a stratigraphic direction or polarity) and orientation data (X,Y,Z, type [contact orientation; fault orientation; fold hinge; etc] a vector; usually expressed as dip direction/dip and the lithology it represents). How do you simultaneously normalize coordinates and vectors? What is method for normalization e.g. min-max?

Thanks for your comment. We have added the related sentence in the subsection of "Training and Validation" in our manuscript. Considering the coordinate ranges of the geological dataset can be much varying from each other, we rescale all the structural data and scalar functions to obtain the normalized training samples. This normalization is implemented by applying a global shift to the entire structural model to make it range from zero and one, which would not change its geological structures. Therefore, we assign the scatter points on the same geological interface to the corresponding iso-values of the normalized model when processing the

structural data. In normalizing orientation vector data, we use the cosine similarity to compare the observed structural orientations and the gradients of the computed model for eliminating their amplitude differences.

2) Lines 328 to 329: You do explain some of these metrics elsewhere in the manuscript, but not all e.g. R2S. You may think to supply these explanations as supplementary material.

Thanks for your constructive suggestion. We have supplied the corresponding explanations of the used regression metrics as supplementary material in Appendix A. To verify the modeling performance of our CNN, we quantitatively measure the differences between the ground-truth structural models and predictions by using various regression metrics including SSIM (Structural Similarity), MSE (Mean Square Error), MAE (Mean Absolute Error), EVS (Explained Variance Score), MSLE (Mean Square Logarithm Error), MDAE (Median Absolute Error), and R Square Score ($R^2S$) in the validation dataset. MSLE measures the prediction performance that corresponds to the expected value of the squared logarithmic error. MDAE is computed by taking the median of all absolute differences and thus can be robust to outliers. EVS is used to measure the proportion of the variability of the predictions in a machine learning method, and the score value can range from zero to one. Higher EVS typically indicates a stronger strength of association between regression targets and predictions and thus represents better network performance. $R^2S$, also called the coefficient of determination, offers a measure of how well the predictions of the network are based on the proportion of total variation. $R^2S$ is similar to EVS, with the notable difference that it can account for systematic offsets in the predictions. In addition, EVS and R2S can be more robust and informative than MAE and MSE in regression analysis evaluation because the former can be represented as percentage errors.

3) Line 339 to 346: This paragraph belongs in the discussion.

Thanks for your suggestion. We have placed this paragraph into the section "Discussion" of the manuscript.

4) Line 348 to 350: The context of this problem isn't clear. Annotate with what, and do other applications do this, but poorly, or not do it at all? What do you mean horizon values – it would seem that it is the iso-value from the scalar field (or fields, especially if considering faults.). Provide some examples with relevant citations.

Thanks for your suggestion. We have modified the related texts in the subsection "Horizon Annotation Experiment" of our manuscript. When training the network, the horizon points along the geological interfaces are assigned to the corresponding iso-values of the implicit model, which is not always the case for real-world modeling applications. Although the annotation of fault points is straightforward by assigning ones on faults and zeros elsewhere, how to annotate appropriate values on the horizon points in real-world applications still remains a problem as the ground-truth model is inaccessible before computation. Therefore, we implement a

data experiment using the horizons with varying annotations and the same faults to study how the different values on the horizon data impact the modeling results of our CNN. By visual comparison in Figure 8, the modeling results are nearly identical to each other, which indicates that our method is not sensitive to the annotations of the input horizon data, which is what we expect. Based on this observation, we recommend to assign scattered points on each horizon to the average vertical coordinate. It is worth noting that the horizon annotations require to be normalized by the model size for consistency with the training dataset.

- **Application:**

1) Section 5.1: Provide a sentence describing the geological scenario and location of study area #1.

   Thanks for your comment. We have added the corresponding sentences in the subsection "Real World 2-D Case Studies" of our manuscript. In this subsection, we apply the trained CNN to a field 2-D seismic dataset to verify its modeling performance for the incomplete structural interpretations with geometrical patterns distinct from the training data. The input structural data are manually interpreted from the seismic images that are randomly extracted from the Westcam dataset. This dataset is acquired in regions with closely spaced and complexly crossing faults with large slips, in which the seismic images are of low resolution due to insufficient coverage.

2) Section 5.1: 2d case studies are simple and not a great test as the horizons are laterally continuous being essentially flat. The fault displacements seem to be honored, which is good. But existing model packages are very effective at producing models with similar simple geometry and topology.

   Thanks for your comment. We have added the corresponding sentences in the subsection "Real World 2-D Case Studies" of our manuscript. In this case study, we apply the trained CNN to a field 2-D seismic dataset to verify its modeling ability in the incomplete structural interpretations with geometrical patterns distinct from the training data. The ambiguous seismic reflections shown in Figure 9a are difficult to be continuously tracked, which causes the noisy and partially missing horizon data displayed in Figure 9b. The faults shown in Figure 9c also might not be fully detected because of the seismic data-incoherent noise and the stratigraphic features that are apparent to structural discontinuities. Moreover, the structural contradictions and hard-to-reconcile features in the structural data might negatively impact the modeling performance of the implicit methods. Therefore, there still remains a challenging task for many traditional approaches to obtain a geologically reasonable model that is structurally consistent with the inputs.

3) Line 401: how do you define complex geological settings here? Polydeformed beds? Domains with vastly different tectonic histories? Multiple magmatic intrusion and extrusion events? Overturned bedding?

Thanks for your constructive comment. We have modified the related words to improve reader's understanding to the content of the complex geological settings in the paragraph. The first seismic data sampled in regions with complexly deformed structures have relatively low resolution and signal-to-noise ratio. As is shown in Figure 11a, some seismic reflections are noisy and difficult to be continuously tracked across the entire volume of interest. The closely spaced and curved faults further complicate the structures especially when there exists data-incoherent noise and stratigraphic features that are similar to structural discontinuities. Therefore, it is challenging for the existing approaches to accurately model the structures from the noisy and partially missing horizons and faults.

4) Figs 11 and 12 have the necessary content, but are too small. Enlarge each panel in the figure, and use more space in the manuscript. Annotations would also help the reader, such as pointing out where the complex parts of the model are (see previous comment), where "the seismic reflections are partially ambiguous and difficult to be continuously tracked" and "results shown in Figure 11c demonstrate that our CNN architecture is beneficial for 3-D structural modeling by predicting a geologically valid model".

Thanks for reminding us of this important point. We guess this question is similar to your Question 3). We have enlarged each panel of the figures and added some annotations in the corresponding figure captions to point out where the complex part of the geological models and seismic structures are for improving the reader's comfort and ease his/her understanding of the content in the subsection "Real World 3-D Case Studies" of our manuscript.

● **Discussion:**
1) Section 6.1 and Fig 13: I like the uncertainty analysis. One thing worth pointing out is that you deal with both aleatory uncertainty (uncertainties resulting from measurement error) and epistemic uncertainty (relating to missing knowledge or data) citing Pirot et al 2022 in GMD. You remove/add drill holes in your fig 13, which is a nice test, so it is worth highlight that point here, or at the very least in the figure caption.

Thanks for your comment. We have modified the texts and added the related reference citation in the subsection "Structural Uncertainty Analysis" of the manuscript. When modeling complex geological structures, the reliability of the implicit methods is heavily dependent on the quality and availability of the input structural data. However, the heterogeneously distributed inputs might cause an ill-posed problem that multiple plausible structural models exist to equally fit the inputs. Therefore, data uncertainty analysis is necessarily critical to looking for an optimal solution, especially for the noisy and hard-to-reconcile structural observations. Although the existing implicit methods can generate various models by perturbing the inputs to characterize uncertainties, they might not explore a broad range of possible geological patterns and structural relationships in nature by using a single model suit for stochastic simulation. Working on the automating of

modeling workflow, our CNN is beneficial for a flexible interpretation of aleatory and epistemic uncertainties by generating diverse modeling realizations instead of one best due to its high computational efficiency.

2) Lines 484 to 487: This is good to acknowledge as modelling these types of structures realistically is challenging, however you can elaborate on why your method is unable to replicate them. I would think the type of geological object is arbitrary given the framework you developed. Is it not possible because the synthetic model generation of Wu et al 2020 cannot recreate unconformities and intrusions? A possible solution maybe to expand the training set to a wider range of objects? Another source could be the Noddyverse (Jessell et al .2022) where dykes, plugs and unconformities are represented. Perhaps you can comment in whether this would work of not. It could be that because you're using a scalar field in the tradition of Lajuanie et al 1997, in which intrusions and unconformities (as distinct from onlap relationships) have not yet been solved (at least to my knowledge).

Thanks for your constructive comment. We have added the related discussions in the subsection "Current Limitations and Improvements" of the manuscript. As structural modeling is dependent on the analysis of the spatial relations of the observed structures to interpolate new geologically valid structures elsewhere, acquiring representative example data is essential for training the CNN to achieve its reliable generalization ability. Therefore, we adopt an automatic workflow to generate diverse models with realistic structures and simulate partially missing horizons. Although working well to recover faulted and folded structures, our CNN might not represent other geological structures not considered in our training dataset, such as unconformities and igneous intrusions. It also might not correctly construct low dip-angle thrust faults in predicted models because we still do not include this type of fault in the currently used data generator. Considering the used training dataset is still not sufficiently large to train a 3-D deep network, future works will focus on the expansion of the training dataset to a broad range of geological geometries and relationships. For example, we can further complicate the simulation workflow by adding more complex and diverse features in the structural models, or using a recently developed open-access 3-D modeling dataset where dykes, plugs, and unconformities are represented.

- **Code and data availability:**
1) I attempted to test the github code on a Windows operating system and had a few issues with conflicting dependencies with installed modules. In particular, I received open MPI errors: "OMP: Error #15: Initializing libiomp5md.dll, but found libiomp5md.dlalready initialized." In attempting to fix this, a number of other dependencies now no longer work.

Thanks for your comment. We suggest using Anaconda and Jupyter Notebook to run our code package on a Windows operating system. All the dependent libraries can be easily installed via "pip install -r requirments.txt" by flowing the instruction.

---

## Author Comment (AC2)

To reviewer 2:

We would like to thank all your wonderful work so that we could get the reviewed manuscript promptly. We appreciate your time in reviewing the work and providing your insightful feedback, which has certainly helped improve the quality and clarity of our manuscript. In this document, we try to address your comments in detail. Let's discuss more if some of our explanations in the responses are not clear to you. The related modifications are not shown in the responses but are all marked in the manuscript revision history.

Thanks!

● **Issues/Suggestions/Questions:**

1) The deep learning architecture is trained using a 2D network (2D convolutions): Section 2.1, P13 L267, P24 L489-491. This is a flaw for 3D applications. The biggest issue is that applying the 2D trained network on a very sparse 3D dataset. 2D slices of your 3D grid must have sampled data points otherwise input to the network is all zero (your binary mask has all zero elements), yielding a scalar field with all zeros. This greatly reduces its applicability for 3D real-world scatter datasets. The presented 3D "real-world" case study had interpretations that covered the modeling domain such that each 2D slice used for inference had sampled data points. To capture geometrical characteristics of 3D geology, 3D convolutions must be employed.

Thanks for your comments. Instead of applying the method to each 2-D slice of very sparse 3-D structural data, we implement a 3-D CNN architecture that constructs a volumetric model from the unevenly distributed scattered horizon and fault points interpreted from geological field observations. We have modified the corresponding words, figures, and captions that might confuse readers in the subsection of "Network Architecture" to improve their understanding to the solutions of the 3-D structural modeling. Although the used 2-D and 3-D CNNs share the same network architecture, the spatial dimensions of the kernel functions are different in their convolutional and pooling layers. As is shown in Figure 2a, we use square brackets to represent the expansion of the corresponding 2-D network to three-dimensional space. To visualize the CNN-based modeling process, Figure 3 displays the hidden representations at each resolution scale of the 3-D CNN that the structural data are passed through. In the subsection of "Real World 3-D Case Studies" of the manuscript, we train a 3-D network with the same architecture to the 2-D network by using the automatically generated synthetic dataset to capture geometrical characteristics of 3-D geology. To demonstrate the performance of our method, we use the trained CNN to construct full models from unevenly distributed structural points sampled from the two totally different field surveys. As shown in Figure 11 and 12, the geologically valid and structurally

consistent results verify the modeling capacities of the CNN in representing complex geological structures.

2) The manuscript shows this approach produces good modelling results for applications where there are significant interpretations. The inputted interpretations provide densely sampled points along horizons forcing strong lateral continuity. How would the trained CNN generalize on 'real world' field observation datasets that are very heterogenous? E.g., sampled points are sparse (large varying distances/gaps between points on the same interface) is some regions along with possible dense sampling from exposed outcrop or borehole in localized parts. It is claimed in the discussion "It is a significant reason why our network could be applied to the real-world datasets acquired in different geological surveys with distinct structural patterns". This is not substantiated. Most geological survey real world datasets are noisy and very heterogenous, this type of data hasn't been used. Interpretative datasets with near continuous sampling along interfaces (horizons/faults) were used. Manuscript would provide larger impact if it can be shown it can be applied to noisy and very heterogenous 3D datasets.

Thanks for your constructive suggestion. We have modified the related texts and discussions in the subsection "Real World 3-D Case Studies" and changed to use more challenging structural data in the first 3-D field data application to demonstrate the proposed method can achieve its reliable generalization in real-world applications. As is displayed in Figure 11, the heterogeneously sampled scatted points are sparse in some localized regions because of the large varying distances among points occur around the geological interface. Based on the automating of data generation workflow, we train a 3-D CNN to correctly capture the geometrical characteristics of 3-D geology. We adopt the trained network in the 3-D field seismic datasets obtained from the totally different surveys to validate its modeling capability. The modeling results shown in Figure 11c demonstrate our CNN architecture is beneficial for 3-D structural modeling and can produce a geologically valid and structurally consistent model. The predicted structural models even maintain the variations of the folded layer structures (highlighted by arrows in Figure 11c and 11d) without global plunge information used to constrain the modeling process. By visual comparison, Figure 11c shows that a group of horizon points sampled at the same geological layer can be accurately located on the corresponding iso-surface of the model, which again demonstrates an excellent fitting characteristic of our network. Additionally, the deep neural network, after training, can be highly efficient to achieve a real-time 3-D structural modeling by using parallel computation in the current GPU platform.

3) Fig. 2 Diverges from how UNet (encoder/decoder) architecture is normally illustrated and makes it confusing. Update so that readers can understand the flow of representations and the figure in general. Clearly indicate tensor dimensionalities. The bracket [x256] I presume means for 3D convolutions which are not employed in any results so why is it here? Expansion mean increasing

dimension of representations by 1x1 convolutions right? why is 1.5? Show where the bottleneck is, since you refer to it in the text.

Thanks for your suggestion. We guess this question is similar to your Question 1). We have modified the corresponding texts and figures to improve the reader's comfort and ease his/her understanding of the content to the 2-D and 3-D network architectures of our CNN. We adopt a linear bottleneck and inverted residual architecture In the encoder branch to make an efficient convolutional structure by leveraging the low-rank nature of the structural interpolation in each learning unit. This structure is composed of an expansion convolutional layer, a depth-wise convolutional layer, and another projection convolutional layer, and each convolution is followed by a Batch Normalization and a Rectified Linear Unit. The two convolutional layers at the ends of the depth-wise convolutional layer are designed to expand the input features to higher-dimensional feature space (one and a half times of the input channels) and project them back to the output channels, such that the block forms a compact feature embedding to improve the expressiveness of the nonlinear transformation at each channel. We did not try a larger expansion factor because of the GPU memory limitation and computation efficiency consideration, but we would suggest to choose a larger size if the GPU memory is allowed.

4) How scattered points are sampled from voxel/pixel grids, e.g., jittering method, how do the 7x7 patches work, how 2 nd order scalar gradients are computed from the voxel/pixels. Requires figure.

Thanks for your comments.
1.    How scattered points are sampled from voxel/pixel grids?
We have modified the related contents to improve readers' understanding to how scattered points are sampled from voxel grids. The input structural data of our network consists of scattered points that are gridded into a volumetric mesh with valid annotations (larger than zeros) on structures and zeros elsewhere. As is displayed in Figure 4b and Figure 5b, we label the points near the faults within one pixel to ones and zeros elsewhere to obtain the input fault data. To obtain the input horizon data, we set the points near the horizon surfaces within one pixel to the corresponding iso-values of the structural model, which ensures the points on the same horizon have consistent annotations. Furthermore, we randomly remove some points from the horizon data in each run of the data generation to simulate the unevenly distributed horizon interpretations in field geological surveys. As the geological interfaces are implicitly embedded in the scalar field with the iso-values and can be obtained by iso-surface extraction methods, we adopt Jittered sampling to randomly choose iso-values and compute their horizons. Specifically, we first divide all the iso-values into uniformly spaced intervals in descending order and then randomly choose one within every interval to extract the corresponding horizons, such that the extracted horizons can be varying instead of being spaced closely. To remove some points from the horizon data, the simplest way is to

randomly generate many square patches and mask the scattered points within the patches, but it might negatively impact the CNN to be well generalized in real-world applications as the inaccessible regions are unlikely in the shape of squares. To solve this issue, we randomize this process by randomly removing some segments from an individual horizon to prevent the network from learning a specific pattern that all the horizon data are partially missing in the same square regions. Specifically, we first sort the points on this horizon into groups in descending order according to their vertical coordinates, and then randomly mask out the points in one or more groups to generate the incomplete data.

2. How do the 7x7 patches work?

In training the CNN, we point-wisely crop square patches from the reference and processed models being measured and compute the loss function within each patch, in which we empirically set every dimensional size of the patch to 7. In this study, all the parameters are selected according to many prior numerical experiments for better modeling performance of our method and kept fixed throughout the study. Although we cannot make sure the used parameter combination is the best one, further parameter tuning is much more time-consuming for a 3-D deep network but hardly obtain further improvements.

3. How 2-nd order scalar gradients are computed from the voxel/pixels?

We have added the corresponding texts and reference citations in the subsection "". Within the deep learning architecture, we can make full use of the various type of geological information in modeling process. For example, we can use structural angles that represent local orientations of geological layers to permit geometrical relationships in the gradient of the scalar function to be considered. The orientation loss is aimed to measure the structural orientation errors between the directional derivatives of the predicted model and the field observations. We adopt the second-order accurate central differences method using the Taylor series approximation to estimate the local orientation at each interior point of the structural model being measured.

5) To produce a model for scattered dataset, there is a preprocessing step that requires the associated scattered points to grid cells to obtain the mask m. Correct? If so, it's important to mention.

Thanks for your constructive comment. We have added the related texts in a new subsection "Structural Data Preprocessing" to demonstrate the preprocessing step that associates the structural scattered points into the grid cells of the model.

In most cases, the structural data collected from field surveys are discrete and not necessarily located on the sampling grid of the model, such that there is a preprocessing step that integrates the structural data into a volumetric mesh with the corresponding annotations.

In our method, we simply shift the horizon and fault interpretations to their nearest sampling grids of the volumetric field and obtain the corresponding scattered points. In the synthetic and the real data applications, the annotation of fault scattered points is straightforward by assigning ones near the faults and zeros

elsewhere. However, although the points along the horizons can be assigned to the corresponding iso-values of the structural model in the synthetic data experiments, this might not be feasible when modeling real-world geological structures. As the ground truth of subsurface structures is typically inaccessible before modeling it, how to properly annotate the horizon scatted points still remains a problem. We thus implement a numerical experiment using the horizons with the different annotations from a simulated model to study how they impact the resultant models of our CNN. As is shown from Figure 8b to 8f, the horizons with different values, together with the same faults, are used as inputs in our network. By visual comparison, the modeling results are nearly identical to each other indicating that our method is not sensitive to the annotations of the inputs, which is what we expect. Based on this observation, we thus recommend assigning the scattered points on each horizon to their average vertical coordinate for correctly following the stratigraphic sequences of geology.

6) While authors have their own limited 3D geological model simulation workflow the following work should be referenced. This could be leveraged for future CNN training, and a potential solution to challenges with training 3D CNN UNet architecture.
Jessell, M., Guo, J., Li, Y., Lindsay, M., Scalzo, R., Giraud, J., Pirot, G., Cripps, E., and Ogarko, V.: Into the Noddyverse: a massive data store of 3D geological models for machine learning and inversion applications, Earth Syst. Sci. Data, 14, 381–392, https://doi.org/10.5194/essd-14-381-2022, 2022.

Thanks for your comment. We have modified the corresponding texts and added the related reference citation in the subsection "Current Limitations and Improvements" of our manuscript. Although working well to recover faulted and folded structures, the proposed method might not represent other geological structures that are not considered in the training dataset, such as unconformities and igneous intrusions. The trained network also might not correctly construct low dip-angle thrust faults in predicted models because we still do not include this type of fault in the currently used data generator. Despite the current limitations, the proposed CNN architecture still shows promising potential to compute a geological valid and structurally consistent model honoring the observed structures. Considering the used training dataset is still not sufficiently large to train a 3-D deep network, future works will focus on further complicating the simulation workflow by adding more complex and diverse geological structural patterns in the models. Considering the used training dataset is still not sufficiently large to train a 3-D deep network, future works will focus on expanding our training dataset to a broader range of geological geometries and relationships. For example, we can further complicate the used simulation workflow by adding more complex and diverse features in the structural models, or adopting a recently developed 3-D geological modeling dataset (Jessell et al., 2022) where dykes, plugs, and unconformities are incorporated.

**● **Detailed Comments/Edits:**

7) P1 L2-L8. All approaches are formulated in terms of a spatial interpolation problem. Within each approach there are numerous mathematical methods to solving the problem. There are very few that setup the problem as a variation problem in which PDEs are solved. Update this part.

Thanks for your advice. We have corrected the "solving PDE" and modified the related words in section "Abstract" of the manuscript. Most advanced implicit approaches formulate structural modeling as least-squares minimization or spatial interpolation problem and use various mathematical methods to solve for a scalar field that optimally fits all the input data under smooth regularization assumption. However, solving the complex mathematical equations with iterative optimization solvers could be computationally expensive in 3-D.

8) P1 L17: Reword. Suggest not using the term 'reasonable geological' until what this means is defined.

Thanks for your comment. We have replaced the term "A reasonable geological model" as "A geological model structurally consistent with subsurface".

9) P1 L20 Incorrect reference. Given reference is for implicit modeling! There is a suite of good reference for explicit methods.

Thanks for your comment. We have corrected the reference citation in this sentence.

10) P1 L20-L21: "It intends"?

Thanks. Corrected.

11) P2 L27-L33: Suggest clearly identifying the advantages over explicit: Fast, reproducible, updatable.

Thanks for your suggestion. Corrected.

12) P2L31-L32: "resultant model from a global respective"?

Thanks, Corrected. We have replaced "respective" with "view" in this sentence.

13) P2 L33: spelling "File"

Thanks, Corrected.

14) P2 L37: "or regional available in"?

Thanks, Corrected. We have removed the phrase "or regional available in" in this sentence.

15) P2 L38: "simplification as structural constraints"? explain.

Thanks for your suggestion. We have added the related interpretations in this paragraph. As it is hardly possible to observe ground truth of subsurface, the geological structures are often heterogeneously sampled in a limited number of

highly developed mining and oil fields. This arises the necessity of adding prior geological rules and assumptions as structural constraints to guide the modeling process. For example, the existing implicit interpolants typically impose explicit smoothness criteria to simplify local variations for computing a unique structural model.

16) P2 L45-47: To deal with this, discontinuous structures are modeled first, which are used to produce a constrained unstructured mesh. Modeling domains are assigned, for which different scalar fields can operate given the geological history. This can work well. Update this.

Thanks for your comment. We have updated the associated discussions in the subsection "Introduction" of our manuscript to demonstrate this problem. Because the scalar function is always continuous on the mesh elements, the mesh elements prohibit from crossing structural discontinuities. Therefore, the conventional methods might not correctly estimate the gradients of the scalar function near the faults or unconformities. To deal with this issue, we typically require to produce a constrained unstructured mesh by carefully modeling the discontinuous structures, such that the DSI methods can still work well in these cases.

17) P3 L65. Incorrect reference de Kemp EA, Jessell MW, Aillères L, Schetselaar EM, Hillier M, Lindsay MD, Brodaric B (2017). Earth model construction in challenging geologic terrain: designing workflows and algorithms. that make sense. In: Proceedings of exploration 17: sixth decennial international conference on mineral exploration, pp 419–439.

Thanks. We have corrected the related reference citation according to your advice.

18) P3 L74. 'sophisticated' remove.

Thanks. Corrected.

19) P3 L75 'is essential for its' remove.

Thanks. Corrected.

20) P4 Fig 3: Unclear why features @ zmax have large activations on encoder branches (there isn't any data there) and is horizontally consistent (same value at ~zmax). How is the output restricted to [0, 1]? You use tanh or softmax activation function in final layer? If so, please provide details.

Thanks for your advice. Instead of using tanh or soft-max activation function in final layer of the network, we apply normalization to hidden representations at each resolution scale of our CNN. We have added the related sentences in this paragraph to explain this problem. Figure 3 visualizes the normalized hidden representations at each resolution scale of our 3-D structural modeling network that the inputs are passed through. Because the amplitude ranges of the hidden representations are much varying from each other, we rescale them to obtain the normalized features with values restrained from zero and one for a visual purpose. The unexpected

activations appearing on the top of the hidden features in the encoder are attributed to the boundary effect caused by operating zero-padding in the convolutional layers. However, as is displayed in Figure 3, the artifacts can be effectively eliminated and corrected in the decoder branch and thus would not negatively impact the resultant modeling quality of our CNN. Also, this is why we need to construct a sufficiently deep network to model geological structures.

21) P4 L80 – 83. Please add: Kirkwood, C., Economou, T., Pugeault, N. et al. Bayesian Deep Learning for Spatial Interpolation in the Presence of Auxiliary Information. Math Geosci 54, 507–531 (2022). https://doi.org/10.1007/s11004-021-09988-0.
Thanks for your suggestion. We have added this reference citation here.

22) P5 L88-89. Method can reproduce results, just not exactly/perfectly.
Thanks for your suggestion. We have modified the related sentences in the manuscript.

23) P6 L125-126 trade accuracy for efficiency?
Thanks for your comment. An optimal trade-off between accuracy and efficiency means that we expect the proposed CNN can produce a model structurally consistent with the inputs while saving computational costs. However, we also would suggest to use a larger network architecture by increasing the expansion factor in each inverse residual module of the encoder branch or doubling the feature channels in the convolutional layers to derive an even better modeling performance if the GPU memory is allowed.

24) P6 L127 regraded (spelling)
Thanks. Corrected.

25) P6 L133 "hierarchical constraints" these are not constraints. They multi-scale features have the capacity to influence structural predictions if the learned multi scale features improve prediction accuracy.
Thanks for your suggestions. We have modified the related texts in the manuscript.

26) P6 L141 "doubles its channels" discrepancy with fig 1 if I interpreted correctly (2->38->76->153->307->614), "linear bottleneck" linear?
Thanks. Corrected. We have replaced "doubles its channels" with "increases its feature channels" in the associated sentences of our manuscript.

27) P6 L142 "low rank nature"?
Thanks for. Corrected. The developed CNN architecture leverages the low-rank nature of the spare and heterogeneously sampled structural data to adaptively suppress uninformative features by using a linear bottleneck and inverted residual structure in each of the encoded convolutional layers. The input structural data have the characteristic of low-rank because they are highly biased with mostly zeros but

only very limited valid labels (annotations) larger than zeros on horizon and fault structures.

28) P6 L147-149 "the block is …"?
Thanks. This block represents the inverse residual block. We have modified the related sentences to improve readers' understanding to our network architecture.

29) P7 L151 spare (spelling)
Thanks. Corrected.

30) P7 L152 "might" From empirical tests, what do the results indicate? In many applications, it is fine to some elements of the features to be zero.
Thanks for your comment. We have modified the corresponding texts in the manuscript. Although the encoder layers aggregate abundant information through recursive channel expansions, not all the features are useful for modeling geological structures. There exist many structurally irrelevant features with mostly zeros across channels due to the spare and heterogeneous characteristics of the inputs. Therefore, treating all channel-wise features equally would waste unnecessary computations to focus on the informative features and thus negatively influences the representational power of the network.

31) P7 L157-159 reference?
Thanks. We have added the corresponding reference citation.

32) P7 L168 "fusing and combining" are the same thing? define MAdds.
Thanks for your comment. We have modified the related texts in the manuscript. By splitting the standard convolution as a two-step process of spatial convolution and channel combination, we can construct a lightweight decoder and effectively reduce the computation complexity and CNN model size. The computational complexity of the method is typically represented as Multiply-Accumulate Operations (MAdds). "MAdds" refers to the total number of multiply and accumulate operations.

33) P7 L173 "non-linear convolutional filters" convolutions are linear transformations.
Thanks for your comment. Our method is designed to progressively complete structural features layer by layer through sequential nonlinear convolutional units that are conditioned on the previous convolutions. Each nonlinear convolutional unit represents a combination of a standard convolutional layer and an activation function that takes the features generated by the convolutional layer and creates the nonlinear activation map as its output.

34) P8 L185 to 189. Indicate the dimensions of m, "points" you mean voxels/pixels? define f, where is y used in the paper?

Thanks for your comment. We have modified the related texts in the subsection "Loss Function" to improve readers' understanding to the notations and formal definitions used in the loss function. In this paragraph, we introduce notations and formal definitions used in this loss function. **x** represents reference structural model, and **m** denotes its binary mask where the pixels or voxels (2-D or 3-D) on the input horizons are set to ones and the rests are set to zeros. We keep the dimensional sizes of the reference model and binary mask consistent with the samples in our training dataset. In addition, $f_\theta$ represents the trained CNN model with trainable parameters $\theta$. We denote the predicted structural model that is replaced with the inputs on the points of the horizon data as **y**.

35) P9 L197 reword. N represents the total number of points within a patch p?
Thanks for your comment. In this sentence, N represents the total number of points within a patch and we crop the patches from the same spatial location in the two structural models being compared.

36) P10 L210 define the gaussian filter
Thanks. Corrected.

37) P10 L216 "might" it will influence
Thanks. Corrected.

38) P10 Eqn 3. Instead of fine-tuning one parameter G_sigma_g, many new parameters are added in which you fix according to Wang 2003. They use those parameters for an entirely different application. How the results compare with just fssim.
Thanks for your suggestion. We have modified the corresponding texts in the manuscript. We set the parameter $\gamma$ in the MS-SSIM function according to many numerical experiments. These parameters are set to weight the SSIM losses computed in the different scale levels for computing the final MS-SSIM loss. Although we cannot ensure the used parameter combination is the best one, further parameter tuning is much more time-consuming for training a deep network but hardly obtain further improvements.

39) P11 Eqn 5. So m and p_bold represent the same thing? A patch?
Thanks for your comment. We have modified the related texts to improve readers' understanding to the notations and formal definitions used in the loss function. According to the formal definitions in staring paragraph of the subsection "Loss Function", **m** denotes its binary mask where the pixels or voxels (2-D or 3-D) on the known horizons are set to ones and the rests are set to zeros. **p** represents the patch cropped from the same spatial location from the structural model. **x** represents reference structural model.

40) P11 L241. Unclear on size of each patch to 7 x 7. Need a figure to know the relations between pixels/voxels, sampled points, and patches surrounding data

inputs, jittering, etc. Would "these parameters" require tuning for very different structural settings?

Thanks for your comment. We have reworded the corresponding texts to improve readers' understanding to the notations and formal definitions used in the loss function. We guess this question is similar to your Question 34). In training the CNN, we point-wisely crop square patches from the structural models being measured (2-D or 3-D) and compute the loss within each patch, in which we empirically set the dimensional size of each patch to 7. The total loss is estimated by averaging the losses computed for the whole cropped patches. All the parameters in the loss function are selected based on many numerical experiments and kept fixed throughout the study to avoid the need for tuning. Although we cannot ensure the used parameter combination is the best one, further parameter tuning is much more time-consuming for training a deep CNN but hardly obtain further improvements.

41) P13 L280-281 unclear

Thanks. We have modified the associated sentences in this paragraph to improve readers' understanding to the horizon iso-value selection. With the jittered sampling method, we first sort all the iso-values into a uniformly spaced grid in descending order and then randomly extract one within each grid unit to compute the corresponding horizon. Therefore, the horizons extracted from the structural model can be spatially varying and not spaced closely.

42) P14 Table 2. How many epochs were performed here? Also, negligible differences, since the input constraints are sampled randomly, you can repeat the experiment and other loss functions could come out first/better

Thanks for your comment. As is shown in Figure 6a, we perform 120 epochs and find the training and validation loss curves converge to low levels when the optimization stops. Table 2 shows the average of the quality metrics on the validation dataset. The CNN trained with the hybrid loss function of MS-SSIM and MAE can outperform the others on all the quality metrics even including the quality metrics which we use as cost function to train the network (MSE and MAE).

43) P14 L297 "much varying"?

Thanks. Corrected.

44) P14 L298-299 Re: Normalization. Clearly indicate what is normalized? I assume its scalar values.

Thanks for your suggestion. We have modified the related texts of the normalization in the manuscript. Considering the coordinate ranges of the field geological datasets can be much different from each other, we rescale every structural model to obtain the normalized one that ranges from zero to one. This normalization is implemented by first subtracting the minimum and then dividing the maximum and thus would not change its geological structures. When

normalizing the structural data, we assign the scattered points on the same geological interface to the corresponding iso-values of the normalized model.

45) **P14 L300 "bath" spelling.**
Thanks. Corrected.

46) **P15 L306-307 reword "has captured…". "very few structural data" subjective term. 5,10, 100, 1000 points?**
Thanks. We have modified the corresponding texts to eliminate these subjective terms in the sentences. The training and validation loss curves gradually converge to low levels when the optimization stops after 120 epochs. The convergence of the loss function demonstrates that the CNN has successfully learned representative geometries and relationships of geological structures from the training dataset.

47) **P15 L309 stability in terms of what? From iteration to iteration? HFA would be computational expensive if the points from iso-surfaces are used. What are the details on how it's computed? Are a sub sample of those points used?**
Thanks for your suggestions. We have modified the related texts and added the discussions in the subsection "Training and Validation". We evaluate the modeling accuracy and stability of our network in terms of the perturbations of the input structures created from the same geological model. In this experiment, we randomly choose 100 synthetic models from the validation dataset and run 20 times of the trained network to calculate the MSE and MAE for each model. By using the iso-surface extraction approach, we can exactly align the sampled points of the horizon surfaces to the input horizon data on the same horizontal grid coordinates. In each modeling process, we randomly generate the horizon scattered points to ensure that the input structural data are different from each other even for the same geological model. Additionally, we propose to use Horizon Fitting Error (HFA) as a quality metric to measure the modeling accuracy of every geological interface associated with the input horizon data by computing an average vertical distance along the depth axis between the horizon scattered points and the iso-surfaces extracted from the predicted model. By using the iso-surface extraction approach, we can exactly align the sampled points of the horizon surfaces to the input horizon data on the same horizontal grid coordinates.

48) **P15 L311 provide the details on the validation dataset (2400 * 0.1). Why not use the entire validation dataset instead of running 20 x of the random sample of 240? Why compute MSE since it is mentioned (and is correct) that it's not a useful error metric (over bias outliers, etc)?**
Thanks for your comments.
1. Why not use the entire validation dataset instead of running 20 x of the random sample of 240?

In this experiment, we evaluate the modeling stability of our CNN in terms of the perturbations of the input structures created from the same geological model. On the one hand, we randomly choose a subset of synthetic models from the validation dataset and run many times of the trained network to calculate the MSE and MAE for each model. On the other hand, we also randomly generate the horizon scattered points to ensure that the input data are different from each other even for the same structural model in each modeling process. Therefore, they are sufficient to demonstrate the great modeling stability of our approach. Additionally, we also show the modeling results and discuss the prediction accuracy of the trained CNN by using the validation dataset in the following subsection "Synthetic Data Examples".

2.  Why compute MSE since it is mentioned (and is correct) that it's not a useful error metric (over bias outliers, etc)?

Although the CNN trained by using MAE and MSE might not correctly guide the network to capture geometrical features whereas blurring high-frequency and sharp discontinuous structures as is discussed in the subsection "Loss Function", they are still appropriate quality metrics that measure the differences between the two structural models being compared.

49) P15 L312-313 unclear.
Thanks. We have reworded the corresponding sentence.

50) P15 L320 "complexly".
Thanks. Corrected.

51) P16 Fig 8. The scalar value differences (0.3) between interfaces are the same. What happens when the differences change? I presume that as delta between scalar values adjacent interfaces (ds) gets larger, accuracy is improved. While as ds gets smaller, accuracy is decreased. If 50 horizons are modelled ds between adjacent surfaces is ~0.02. Does ds impose/bias constant unit thickness between interfaces?
Thanks for your comment. We have added the new experiments and modified the corresponding texts in the newly added subsection "Structural Data Preprocessing". As the ground truth of geological structures is typically inaccessible before modeling, how to properly annotate the interpreted horizons still remains a problem. We have changed the numerical experiment and utilized the horizon data labeled with different iso-values in a synthetic structural model (Figure 8a) to study how they impact the predictions of our CNN. In this experiment, the scattered points on two horizons are assigned by the normalized iso-values that range from 0.3 to 0.8 with three distinct intervals of 0.2, 0.3, and 0.4. As shown from Figure 8b to 8f, the network takes the horizon data with various iso-values and the same fault data to produce full structural models as outputs. By visual comparison, the nearly identical modeling results indicate that the method is not sensitive to the different data annotations within a reasonable range, which is what we expect. Additionally, we can observe a larger interval of the horizon annotations is contributed to a more

significant displacement of geological layers on the opposite of the fault structures in the predicted model (Figure 8c and 8e). Based on this observation, we recommend to label the scattered points on each horizon with their average vertical coordinate for correctly following the stratigraphic sequences of geology. Note that the annotations on horizons require to be consistently rescaled by the model size to keep consistent with the normalized training dataset.

52) P16 L348 "scatted" spelling
Thanks. Corrected.

53) P17 L369 "partially missing horizon" a tiny % is missing, "might not be"? are they or aren't they fully annotated?
Thanks for your comment. We have modified the corresponding texts to better demonstrate the potential challenges of obtaining a geologically reasonable and structurally consistent model. As is shown in Figure 9a, the ambiguous reflections are difficult to be continuously tracked across the entire seismic images, which causes the partially missing horizon data shown by different colors in Figure 9b. In addition, not all the faults are detected from the seismic images because of data-incoherent noise and stratigraphic features apparent to discontinuous structures. Moreover, the structural contradictions and hard-to-reconcile features in the inputs might negatively impact the modeling quality of geological structures.

54) P18 L372-373. For the datasets used in the manuscript, traditional methods can make good geological models consistent with the inputs.
Thanks for your suggestion. We guess this question is similar to your Question 2). We have changed the input interpretation data of the first 3-D field data application and modified the related texts in the subsection "Real World 3-D Case Studies". As is shown in Figure 11, the heterogeneously sampled scatted points are sparse or clustered in some localized regions because of the large variations of the distances among the points occur along the same geological interface. The modeling results shown in Figure 11c demonstrate that the CNN architecture is beneficial for 3-D structural modeling by predicting a geologically valid model, where the structural discontinuities and the interfaces are consistent with the given scattered points. The predicted models even maintain the variations of the folded layer structures (highlighted by arrows in Figure 11c and 11d) without global plunge information used to constrain the modeling process. By visual comparison, Figure 11c shows that a group of horizon points sampled at the same geological layer can be accurately located on the corresponding iso-surface of the model, which again demonstrates a great fitting characteristic of our network.

55) P21 Fig 13. "Geological uncertainty analysis" Multiple realizations are made given different interpretations. What "analysis" was done? How is uncertainty here quantified?

Thanks for your suggestion. We have changed the subtitle as "Structural Uncertainty Characterization" and modified the corresponding contents in this subsection of our manuscript. The heterogeneously distributed structural data pose an ill-posed problem that there exist multiple plausible structural models which equally fit the inputs. For this reason, data uncertainty analysis is necessarily critical to looking for an optimal solution, especially for the noisy and hard-to-reconcile structural observations and interpretations. Although the existing implicit methods can generate various models by perturbing the inputs to characterize uncertainties, they might not explore a broad range of possible geological patterns and structural relationships in nature by using a single model suit for stochastic simulation. Working on the automating of modeling workflow, our CNN is beneficial for a flexible interpretation of aleatory and epistemic uncertainties by generating diverse modeling realizations instead of one best due to its high computational efficiency. We use various combinations of modeling objects with the horizons and faults interpreted from the borehole and the outcrop observations to study the uncertainties associated with the position variations of geological structures. The simplest possible structural model consists of multiple continuous and conformal horizons in the first data example. By contrast, the modeling situations become more complex when considering additional geometrical objects of faults that dislocate the geological layers. In addition, we randomly perturb horizon positions to yield the variations in layer thickness because the stratigraphic interface transition might not be accurately observed from the vertical boreholes. These structural data are used as inputs of the network in modeling geological structures to demonstrate the proof of concept.

56) P21 L428-429 And relies on the availability of knowledge and how those methods incorporate that knowledge particularly for data sparsity increases.

Thanks for your comment. We formulate implicit modeling as image inpainting with deep learning, in which a full structural model is estimated from the sparse and heterogeneously sampled data based on the past experiences and knowledge learned from a sufficiently large training dataset. That also means, when the network is trained well, the modeling experiences and knowledge learned from the synthetic dataset are implicitly embedded in the CNN model parameters. This characteristic permits a flexible introduction of empirical geometrical relations and structural interpolation constraints by defining an appropriate loss function to measure the structural differences between the CNN predictions and the reference models. Our network can produce a structural scalar field as an implicit representation of all the structures from various types of the structural data such as horizons that encode the stratigraphic sequence of the sampled interfaces, and faults in the presence of the geological boundaries.

57) P21 L432-434. Given a scattered data set of geological observations, the trained CNN produces 1 model. It doesn't generate a "diverse [set] of possible modeling realization" ([] grammar fix). Fig 13 shows given different interpretative

inputs, different realizations are produced. Generating interpretative inputs is extremely time consuming, so the point that the CNN are very computational efficient for generating ensemble realizations is incorrect until the method can generate those realizations without interpretative inputs.

Thanks for your comments. Corrected. We guess this question is similar to your Question 55). Working on the automating of modeling workflow, our CNN is beneficial for a flexible interpretation of aleatory and epistemic uncertainties by generating diverse modeling realizations instead of one best due to its high computational efficiency. We can use various combinations of modeling objects with the horizons and faults interpreted from the borehole and the outcrop observations to study the uncertainties associated with the position variations of geological structures. For example, we can randomly perturb positions of interpreted interfaces to yield the variations in layer thickness as the stratigraphic interface transition might not be accurately observed from the vertical boreholes. These structural data are used as a diverse set of inputs of the network for modeling the possible structural geometries and relationships to demonstrate the proof of concept.

58) P23 L453-454 z variable? Predicted model was y_hat before. Show formula for grad y_hat/z. This uses the neighbouring cells surrounding a given pixel/point?

Thanks for your suggestions. We have added the corresponding discussions in the subsection "Structural Orientation Constraint" of the manuscript. The loss function of orientation constraint is proposed to measure the angle errors between the directional derivatives of the predicted structural model and the orientation observations by using cosine similarity. We adopt a second-order accurate central differences method with the Taylor series approximation to estimate the local orientation at each interior point of the given structural model $\mathbf{z}$. We compute the cosine similarity between the normal vector and the orientations in the reference model $\mathbf{x}$ and the predicted model $\mathbf{y}$, respectively. Therefore, the orientation loss function is proposed to measure the structural angle errors between the two models being compared.

59) P23 Eqn 8. You manually tune lamba, beta? Apply MTL principles? What values did you use?

Thanks for your suggestions. We guess this question is similar to your Question 4). We have added the corresponding discussions in the manuscript. $\beta$ and $\lambda$ are empirically set to 1.00 and 1.25, respectively. All the parameters in the loss function are empirically selected according to many prior numerical experiments for better modeling performance of the CNN and kept fixed throughout the study to avoid the need for tuning. Although we cannot make sure the used parameter combination is the best one, further parameter tuning is much more time-consuming for a deep network but hardly obtain further improvements.

60) P24 L489-491 "the used training dataset is still not sufficiently large to train a 3D deep network" To create 3D model, you slice 3D grid into 2D slices, each of which you perform inference on using the trained 2D network? Critical issues here if this is the case.

Thanks for your comments. We guess this question is similar to your Question 1). Our proposed 2-D and 3-D CNNs have the same architecture, and the only difference is their kernel spatial dimensions in the convolutional and pooling layers. As is shown in Figure 2a, we use square brackets to represent the expansion of the corresponding 2-D network to three-dimensional space. Although working well to recover faulted and folded structures, the proposed method might not represent other geological structures that are not considered in the training dataset, such as unconformities and igneous intrusions. Considering the used training dataset is still not sufficiently large to train a 3-D deep network, future works will focus on expanding our training dataset to a broader range of geological geometries and relationships. For example, we can further complicate the used simulation workflow by adding more complex and diverse features in the structural models, or adopting a recently developed 3-D geological modeling dataset where dykes, plugs, and unconformities are incorporated.

61) P25 L497 "spare" spelling.

Thanks. Corrected.

62) P25 L506 "noisy structures" sample interfaces are not noisy. You can clearly see scattered points on sampled interface there is no positional (or orientational) fluctuation in adjacent/nearby points.

Thanks. We have modified the corresponding sentence as "In both synthetic data and real-world data applications, we verify its modeling capacities in representing complex structures with a model geologically reasonable and structurally consistent with the inputs." in the section "Conclusions" of the manuscript.

---

## Author Comment (AC3)

Responses to comments from reviewers

Dear editors,
We sincerely appreciate all your wonderful work so that we could obtain the reviewed manuscript promptly. Thanks for your time on reviewing and processing our paper and providing your insightful feedback. Below we are trying to responses your suggestions and questions. Let's discuss more if some of our explanations in the responses are not clear to you.

Thanks!

- **Code and Data Availability**
1) Please publish your code in one of the appropriate repositories. Also, please, remind that you must include in a potential reviewed version of your manuscript the modified 'Code and Data Availability' section, with the DOI of the code.
   Thanks for your suggestion. We have modified the related texts in section "Code and Data Availability" of the manuscript. The source code for the CNN developed in Pytorch in this research is available and can be accessed through the following DOI link of Zenodo:
   https://doi.org/10.5281/zenodo.6684269
   The synthetic structural models, used for training and validating the network, are uploaded to Zenodo and are freely available through the DOI link:
   https://doi.org/10.5281/zenodo.6480165.

---

## Author Response (AR2)

Responses to comments from reviewers

To reviewer 1:

Dear reviewer,

We sincerely appreciate all your careful reviewing so that we could get the reviewed manuscript promptly. We appreciate all your valuable comments and suggestions, which help a lot to improve our manuscript. Below we are trying to responses all your comments, suggestions, and questions. Thanks!

1) The revised manuscript describes much better that there is indeed two different networks that are trained (one for 2D and one for 3D) using the same architecture. However, the sentence "Considering the used training dataset is still not sufficiently large to train a 3-D deep network" should be reworded. Your current training set is sufficiently large to train a 3D deep network for the geological settings that you have applied them for in the paper. However, to support modeling in more complex settings, expansion of this training set is required to sample these settings so the network can produce good models for these settings on unseen data.

Thanks for your comments. We have modified the corresponding sentence in the section "DISCUSSION" of the manuscript as, "Considering the used training samples are still not sufficiently diverse to support modeling complex and unseen geological settings, future works will focus on expanding the training dataset to a broader range of structural geometries and relationships related to these settings".

2) Fig 11 e) (modeling interfaces) and Fig 12 g) and h) (recovered full horizons). The modeled interfaces extracted using iso-surface extraction methods on resulting scalar fields have gaps in them due to faulting (Fig 11 d) and Fig 12 f). The "recovered full horizons" have gaps filled in, and have characteristic bumps in these locations. Why is this the case? Also the property that the color map is representing on these surfaces were never mentioned, looks like normalized x or y coordinates.?

Thanks for your suggestions. We have added and modified the related texts to demonstrate the structural gaps shown in the modeled interfaces near the faults in subsection "Real World 3-D Case Studies" of the manuscript. The modeling results shown in Figure 11c demonstrate that the CNN architecture is beneficial for 3-D structural modeling by predicting a geologically valid model. We extract the full geological interfaces from the resulting scalar fields by using the iso-surface extraction method and mask the surface segments near the faults to highlight the structural gaps due to faulting in Figure 11d. Figure 11e displays a single modeled interface without masking and colored via vertical coordinates, in which there exist sharp vertical jumps across the faults. As is displayed in Figure 11d and 11e, the modeled structural discontinuities and interfaces can be consistent with the inputs, and the predicted models even maintain the folding structural variations

(highlighted by arrows) without global plunge information used to constrain modeling.

3) P18 L379-382 "method is not sensitive to the different data annotations". Is this an accurate characterization given "Additionally, we can observe that a larger interval of the horizon annotations is contributed to a more significant displacement of geological layers on the opposite of the fault structures in the predicted model (Figure 8c and 8e)" P18 L381-382 and " how to properly annotate the interpreted horizons remains a problem" P18 374-375..

Thanks for your comments. We have modified the corresponding sentences in subsection "Structural Data Preprocessing" of the manuscript to improve readers' understanding to the robustness of the method against the variations of the input data annotations: "By visual comparison, the nearly identical predictions indicate that the modeling accuracy is not sensitive to the changes of horizon annotations within a reasonable range, which is what we expect. Additionally, in comparison of Figure 8c to 8e, we also observe that a larger gap of the horizon annotations causes a more significant displacement of geological layers on the opposite of faults in the predicted model."

4) Fig 8 c) and e). A fault appears to be introduced on the far right hand side of the two sections when there is no data to support that feature. Suggesting that because geological knowledge and relationships are not incorporated explicitly as constraints, the approach may introduce geological features/structures that are not there in reality.?

Thanks. The undesired discontinuous features near the right boundaries of the two sections in Figure 8 are caused by the edge artifacts from the recursive convolutions followed by zero-padding operations in the CNN. Thus, the modeling accuracy and stabilities near the boundaries are less than the model elsewhere. Although existing in almost all the deep learning methods, the edge effect can be well addressed by expanding the model size before and extracting the submodel we are interested in from the final prediction.

5) P26 L531. "complicate the used" suggest "augment" or "enhance".

Thanks for your suggestion. Corrected.

6) P3 L78 "complexly nonlinear spatial relations" ?

Thanks for your suggestion. We have modified the related sentence as "CNN is essential for its remarkable power in analyzing geometrical features and capturing complexly nonlinear mapping relations between the inputs and outputs given a sufficiently large training dataset".

---

## Author Response (AR3)

Responses to comments from editors

Dear editors,

To associate editor:
Thanks for your time in reading and processing our paper. We appreciate all your valuable comments and suggestions, which are helpful to improve our manuscript. We have added the discussions corresponding to #4 in the previous round of reviewing in our manuscript to demonstrate the boundary effects that may introduce instabilities to the modeling results and how to mitigate the edge artifacts in the CNN. Let's discuss more if some of our explanations in the responses are not clear to you.

Thanks!